# Evolution of cortical geometry and its link to function, behaviour and ecology

Ernst Schwartz [1], Karl-Heinz Nenning[1,2], Katja Heuer [3], Nathan Jeffery[4], Ornella C. Bertrand [5,6], Roberto Toro[3], Gregor Kasprian[1], Daniela Prayer [1] & Georg Langs [1,7] ✉

Studies in comparative neuroanatomy and of the fossil record demonstrate the influence of socio-ecological niches on the morphology of the cerebral cortex, but have led to oftentimes conflicting theories about its evolution. Here, we study the relationship between the shape of the cerebral cortex and the topography of its function. We establish a joint geometric representation of the cerebral cortices of ninety species of extant Euarchontoglires, including commonly used experimental model organisms. We show that variability in surface geometry relates to species' ecology and behaviour, independent of overall brain size. Notably, ancestral shape reconstruction of the cortical surface and its change during evolution enables us to trace the evolutionary history of localised cortical expansions, modal segregation of brain function, and their association to behaviour and cognition. We find that individual cortical regions follow different sequences of area increase during evolutionary adaptations to dynamic socio-ecological niches. Anatomical correlates of this sequence of events are still observable in extant species, and relate to their current behaviour and ecology. We decompose the deep evolutionary history of the shape of the human cortical surface into spatially and temporally conscribed components with highly interpretable functional associations, highlighting the importance of considering the evolutionary history of cortical regions when studying their anatomy and function.

In mammals, the cerebral cortex represents the most important anatomical structure for cognition[1]. As such, it is the focal point of a large body of neuroscientific research, investigating its numerous behavioural correlates as well as its exceptional evolutionary expansion and progressively extended ontogeny in humans[2,3]. Despite dramatic evolutionary changes and diversification in the mammalian lineage[4], the organisation of the cerebral cortex retains a range of common underlying attributes. For example, folding wavelength[5], myeloarchitecture[6], as well as structural[7] and functional connectivity[8] are sufficiently similar to establish correspondences across species. These commonalities coincide with a remarkably stable macroscale topography of the mammalian cerebral cortex[9–11] characterised by anatomically and functionally defined cortical areas that evolved and developed under strong genetic[12] and ecological[13] constraints.

However, the relationships between these organisational principles of cortical structure and function are yet poorly understood[14].

[1]Department of Biomedical Imaging and Image-guided Therapy, Computational Imaging Research Lab, Medical University of Vienna, Vienna, Austria. [2]Center for Biomedical Imaging and Neuromodulation, Nathan Kline Institute, Orangeburg, NY, USA. [3]Institut Pasteur, Université Paris Cité, Unité de Neuroanatomie Appliquée et Théorique, F-75015 Paris, France. [4]Institute of Life Course & Medical Sciences, University of Liverpool, Liverpool, England. [5]Institut Català de Paleontologia Miquel Crusafont, Universitat Autònoma de Barcelona Edifici ICTA-ICP, c/ Columnes s/n, Campus de la UAB, 08193 Cerdanyola del Vallès., Barcelona, Spain. [6]School of GeoSciences, University of Edinburgh, Grant Institute, Edinburgh, Scotland EH9 3FE, United Kingdom. [7]Computer Science and Artificial Intelligence Lab, Massachusetts Institute of Technology, Cambridge, MA, USA. ✉e-mail: georg.langs@meduniwien.ac.at

Aside from insights into evolution, understanding the nature and causes of the brains' phenotypic variability is relevant when translating neuroscientific results in model animals to humans. Here, studying the evolutionary history of the brain poses a promising avenue of research[15,16].

The heterogeneous change of the extent and number of individual cortical areas during evolution[17] has spurred a debate on whether these changes of function and topography occurred in a concerted manner[18] or are better described as a mosaic of dynamically changing areas[19]. Comparative studies of both extinct and extant species have been used to study the evolutionary history of the cerebral cortex. Due to taphonomic processes, the estimation of this history with reference to extinct taxa has come to heavily rely on measures derived from cranial endocasts. Such measures of endocranial volume[3], shape[20], or osteological landmarks[21] all represent the outer meningeal surface and thus lack fidelity regarding the true cortical surface. This loss in anatomic precision limits the translation across both extant and extinct taxa and has led to conflicting theories and continued controversy regarding, for example, primate brain evolution[22–24].

The recent availability of large open image datasets of rodents[25], non-human primates[26,27], human subjects[28,29] and fossil taxa[30] has facilitated comparative analyses of brain organisation and function[31,32]. However, studies are either limited to phylogenetically closely related species[33,34], a-priori defined spatial homologies[6,35], or avoid establishing correspondences between morphologically and functionally divergent domains altogether by focusing on global descriptions of brain organisation[5,36–38].

Addressing these shortcomings and adequately studying the distributed patterns and sources of cortical variability and evolution requires a common analytical framework[31] that bridges the gaps between the analysis of humans, extant mammals and the fossil record.

Here, we link phylogenetic and geometric methods to establish a common reference frame for the cerebral cortex of Euarchontoglires, the supra-order containing both humans as well as commonly used neuroscientific model animals such as mice, rats, macaques and marmosets. By establishing group-wise correspondences between all cortical surfaces in a dataset of ninety different species, we show how ecological adaptations are reflected in the shape of the surfaces of cerebral cortices of living members of Euarchontoglires. We use spatial statistics to quantify the process of modal segregation of brain function. Finally, we combine ancestral state reconstruction with statistical decoding of brain activation to derive a historical time-line of modules of evolutionary adaptations of the cortical shape in the deep ancestral human lineage since the last common ancestor of rodents and primates. The resulting aligned surface meshes for all extant species, their coordinates for commonly used template spaces and tools for mapping between them as well as their estimated ancestral shapes are available at https://github.com/cirmuw/EvolutionOfCorticalShape.

## Results

### An evolutionary common reference frame of cortical geometry for comparative neuroscience

We established a common reference frame to map correspondences between densely sampled cortical surface models of ninety extant species of Primates (58), Rodentia (28), Lagomorpha (2), Dermoptera (1) and Scandentia (1) (Fig. 1; Supplementary Data 1a). For this, we first performed manual segmentations of the cortical surfaces on available MRI (75), DiceCT (7) and serial histology (8) data. We computed the convex hull of these segmentations as the simplest covering of the outer surface without any folds or creases. The surface areas of the resulting as well as the original model serve as global features of the cortical geometry. We furthermore collected simplified descriptions of species behavioural ecology in terms of sociality, activity pattern and preferred habitat (Fig. 1; Supplementary Data 2). We obtained a calibrated phylogeny for the analysis from the consensus tree of 100

realisations of a tip-dated Bayesian model of divergence times[39] on which we estimated the ancestral states of the area of the cortical surfaces and their convex hulls[40] as proxies for cortical morphology (Supplementary Figs. 1 and 2). We then performed pairwise matching between the cortical surfaces of sister species, and approximated their ancestral state by interpolating in the space of smooth shells[41,42]. There, we used the relative differences in the global shape features to determine the interpolation factor. The relative position of the estimated ancestral and observed daughter species in the nonlinear space of smooth shells thus corresponds to the euclidean distance between the corresponding global shape features (Fig. 2). We iterated these steps, progressing backwards in the phylogenetic tree. Upon reaching its root node, we resampled all surfaces to a common topology by placing surface vertices at geometrically corresponding locations[43]. Details of the modelling and analytic workflow are provided in the Supplementary Methods. We evaluated the appropriateness of the resulting common reference frame by relating measurements obtained from the inherent group correspondences to established knowledge from evolutionary neuroscience.

### Neuroecology of cortical morphology

The relationship between the morphology of a-priori defined cortical areas and sensory specialisation has been studied extensively in a variety of species, both in their natural state as well as by manipulation such as post-natal enucleation[44]. Using the common cortical reference space, we performed a data-driven analysis to study if the relationship between the morphology of cortical areas and sensory specialisation can be observed at the whole-brain level.

First, we performed phylogenetic principal component analysis (pPCA)[45] to reduce the dimensionality of our dataset and tested for the association between the extracted modes of variation describing the shape of the cerebral cortex and ecological and behavioural niches. We found 5 pPCA components out of 21 (Supplementary Fig. 2) that showed significant ($q < 0.05$, Wilcoxon-Rank-Sum/Mann-Whitney U-Test, FDR corrected) association for any of the observed ecological and behavioural variables of arboreality, terrestriality, fossoriality or large group size. Despite the strong overlap with group size, no component was associated with diurnality. We then aggregated the information contained in the pPCA modes that showed a significant relationship to individual ecological factors (Supplementary Fig. 2). Based on these factors, we estimated the areal differences corresponding to each ecological variable and decoded the resulting expansion maps into neuroscientific concepts (Fig. 3; Supplementary Data 3).

By mapping the relative expansion patterns into a functionally defined parcellation of the human cerebral cortex[46], we found significant differences (Kruskall-Wallis, $p < 0.001$, Supplementary Data 5) in the cortical expansion patterns associated with every investigated ecological and behavioural variable except diurnality. The largest effects were observed for ecological variables of habitat (fossorial: $p < 1e\text{-}6$, $\eta^2 = 0.21$, terrestrial: $p < 1e\text{-}6$, $\eta^2 = 0.14$, arboreal: $p < 1e\text{-}6$, $\eta^2 = 0.18$) whereas large group size showed lower effect ($p < 1e\text{-}6$, $\eta^2 = 0.1$).

Post-hoc tests revealed significantly differing expansion patterns associated with both ecological and behavioural variables. Visual and dorsal attention areas are most strongly affected by habitat, where they are reduced in fossorial and expanded in arboreal species. Their relative expansion is also positively associated with sociality (i.e., large social groups), while limbic areas show a negative relationship (Supplementary Fig. 5). We found a significant correlation between the pairwise effect measures for all variables related to habitat (fossorial:terrestrial: $r(19) = 0.592$, $p = 7.74e\text{-}3$, 95% CI = [0.145, 0.839], fossorial:arboreal: $r(19) = 0.635$, $p = 4.33e\text{-}3$, 95% CI = [0.206, 0.859], terrestrial:arboreal: $r(19) = 0.933$, $p < =1e\text{-}6$, 95% CI = [0.797, 0.979], Supplementary Data 4a) that implied an ordering in terms of habitat complexity (Figs. 4a and 6b). We performed a joint analysis of the

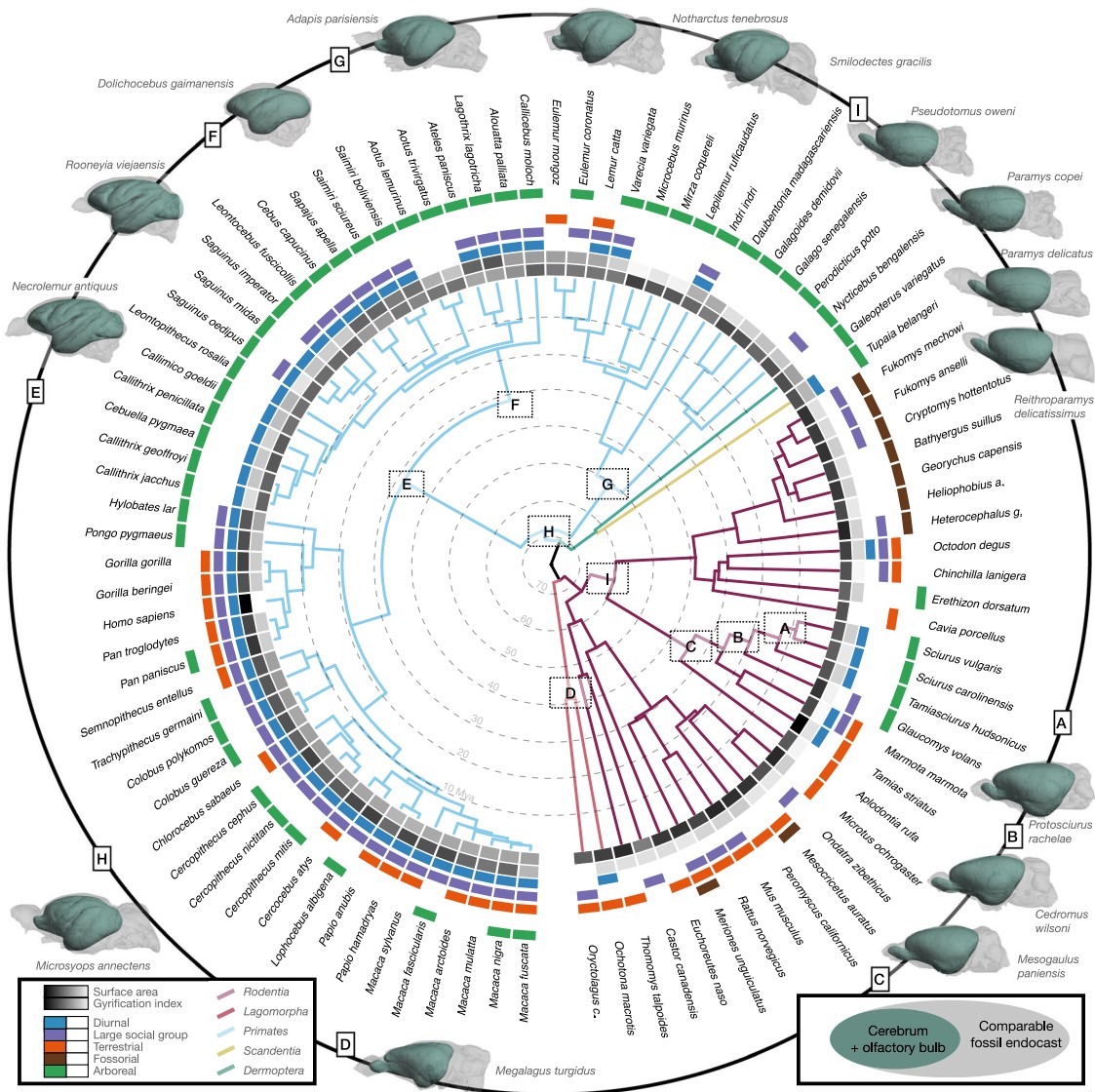

**Fig. 1 | Phylogenetic tree of the dataset used in this study.** The dataset covers 90 species (58 primates, 28 rodents, 2 lagomorphs, 1 scandentia, 1 dermoptera), for which 5 binary ecological and behavioural categorical indicator variables were collected (Large group size, diurnality, arboreality, terrestriality and fossoriality, see Supplementary Datas 1a, 2a and 2b). Estimates of the morphology of potential ancestral shapes were computed after iterative pairwise alignment of sister species. The resulting surface models are compared with endocasts (Supplementary Data 1b) obtained from phylogenetically closely related fossil endocasts at the branching points indicated by the letters A-I, highlighting the morphological plausibility of the results and consequently the underlying surface correspondences.

habitats ordered from arboreal to terrestrial to fossorial species (Fig. 4c; Supplementary Data 4b) that revealed a sequential decrease in relative cortical expansion in visual, motor, frontal parietal as well as default mode related areas. We quantified the consistency of these effects using repeated measurements correlation[47] (Supplementary Data 4c), which were significant in all but the ventral attention areas ($r_{rm}(241) = -0.048$, $p = 0.45$, 95% CI [−0.174, 0.078]), whereas visual ($r_{rm}(323) = -0.802$, $p < 0.0001$, 95% CI [−0.837, −0.759]) and dorsal attention regions ($r_{rm}(243) = -0.762$, $p < 0.0001$, 95% CI [−0.810, −0.704]) were most strongly affected by differences in habitat complexity. Together, these results show that the morphology of the cerebral cortex reflects ecological and behavioural factors that pose specific cognitive demands to individual species[48].

## Evolution of the topography of modality-specific functional areas in the cerebral cortex

Evolutionary adaptation to specific environmental niches and neuroplasticity of the cerebral cortex cause individual cortical areas to become more attuned to specific types of sensory processing[49,50]. In extant species, this adaptation was successful and resulted in increases in overall body and brain size. The combination of these effects leads to an expansion of cross-modal processing regions and a segregation of sensory processing regions[51].

In order to test if the proposed common reference frame reflects this cortical organising principle, we projected a map of cortical sensory specialisation for auditory, visual and somatosensory processing of individual brain regions[52] onto the common space to estimate the functional modal specialisation of cortical areas in the rodent-primate ancestor (Fig. 5). Using the cosine distance between sensory specialisation measures we then performed semivariogram modelling with respect to geodesic distance on all cortical surfaces. In spatial statistics, the largest distance at which two measurements are spatially autocorrelated is called the range of the model. A decreasing range value corresponds to increasing segregation into modality specific areas (Supplementary Fig. 9).

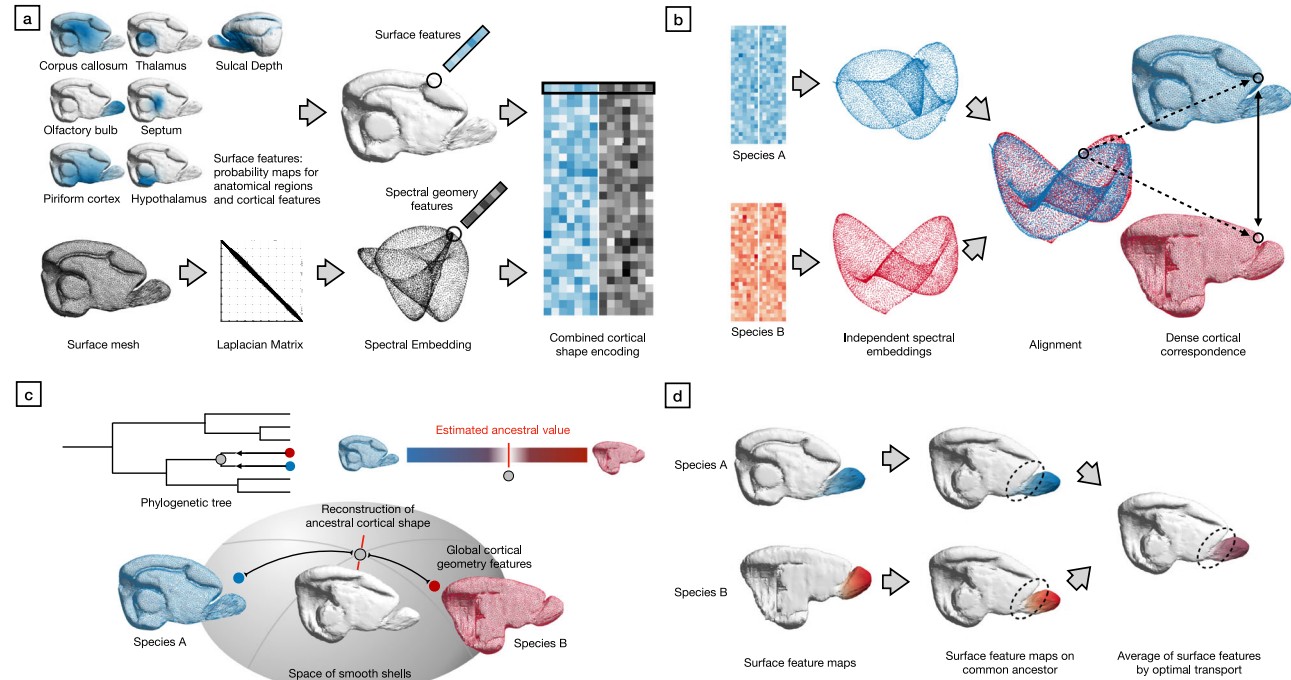

**Fig. 2 | Reconstructing the cortical shape of a common ancestor.** Overview of the steps required to perform shape alignment and ancestral state reconstruction between the surface models of the cerebral cortices of two sister species. **a** For two species, anatomy and geometric features are assigned to each point on their cortical mesh; (**b**) after spectral embedding of both surfaces, alignment establishes dense correspondence between the cortices of the two species; (**c**) the shape of the common ancestors is reconstructed based on these correspondences by interpolating in the space of smooth shells; (**d**) finally, cortical features are estimated for the common ancestor, and the reconstruction is iterated until the root of the phylogenetic tree is reached.

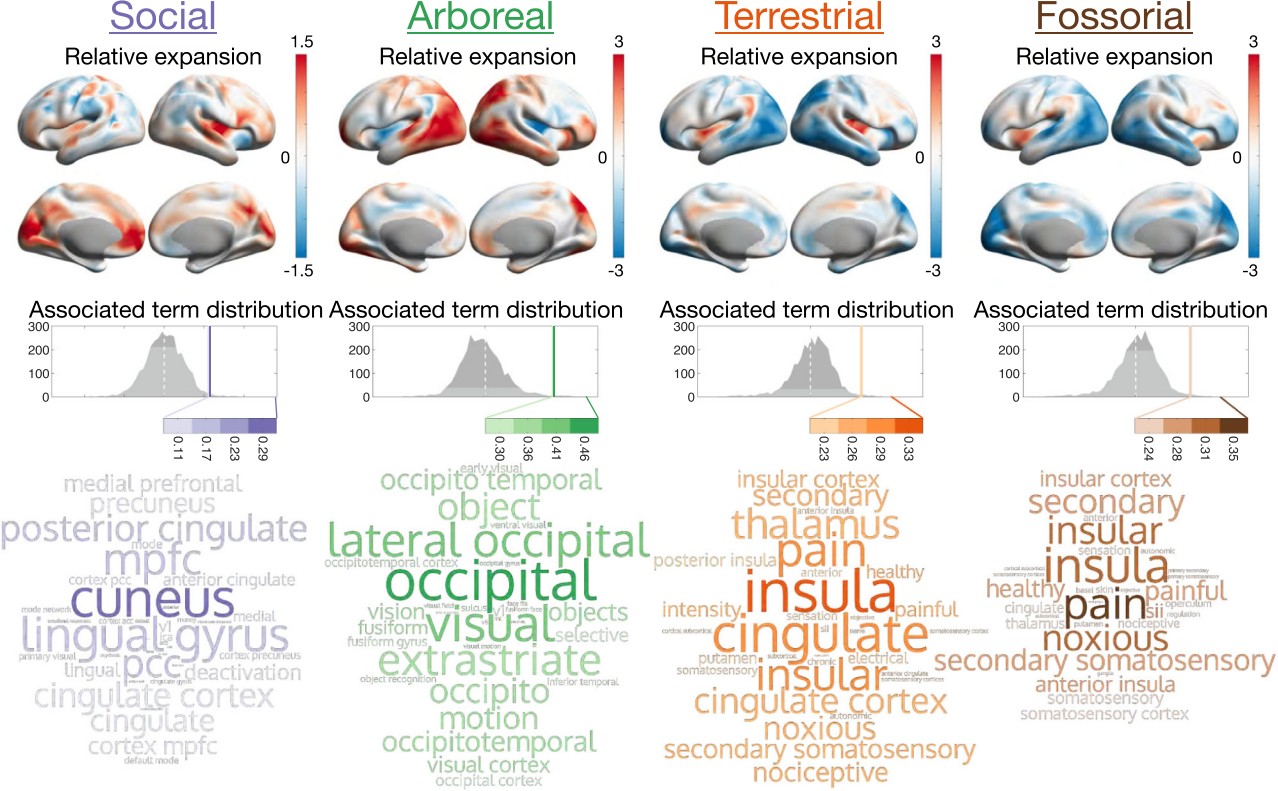

**Fig. 3 | Relative expansion along ecological dimensions reveals functionally associated areal patterns across the cortex.** The changes in surface area between the extremes of synthetically generated linear combinations of pPCA modes associated with individual ecological variables are subjected to decoding via the NeuroSynth database. Correlation of the expansion maps and individual terms are colour-coded for each socio-ecological variable. Retaining the 99th percentile of most strongly correlated terms (Supplementary Data 3) indicate that these expansion maps encode related, semantically meaningful concepts, thus highlighting the association of evolutionary adaptive processes in both behavioural and ecological variables and cortical morphology.

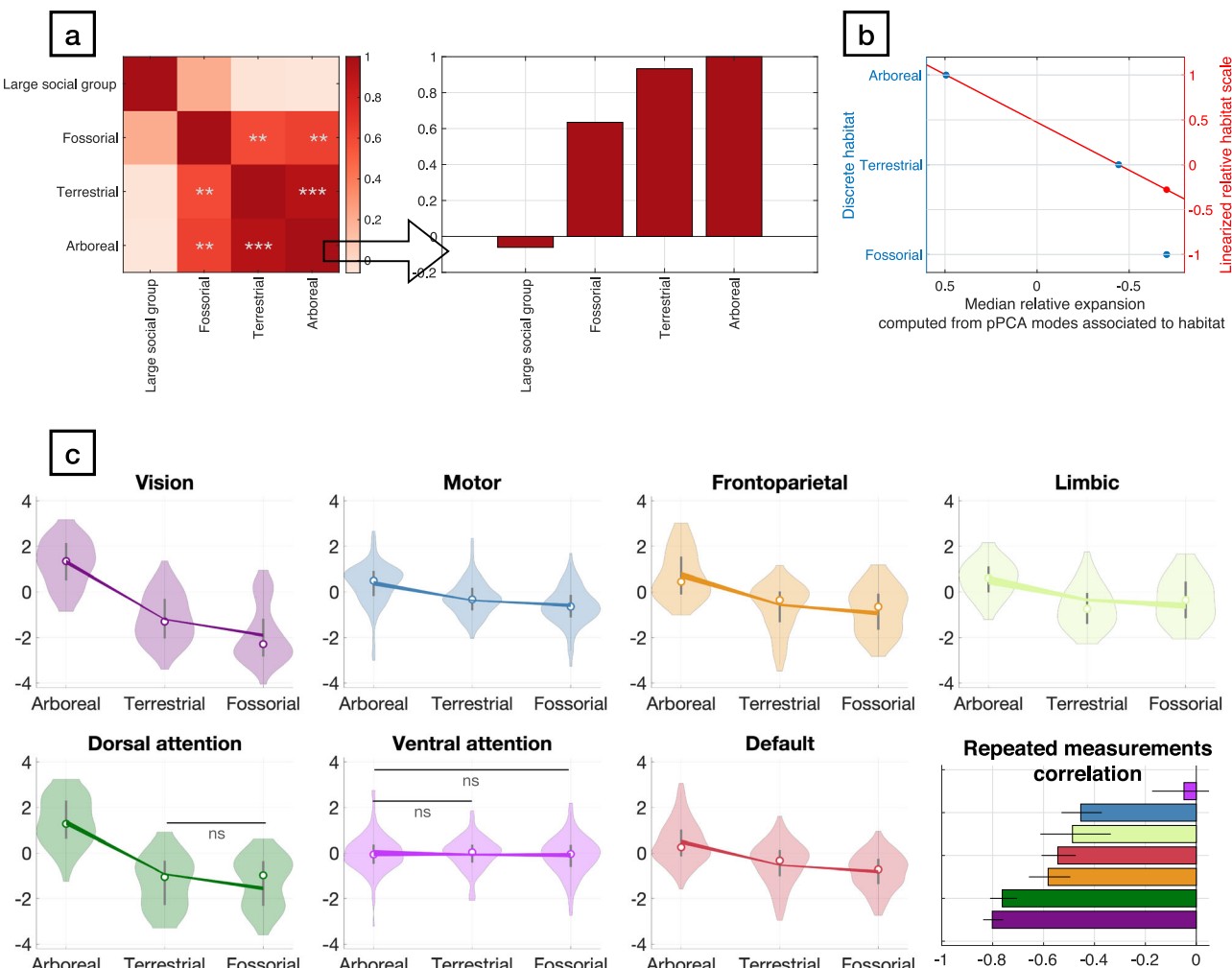

**Fig. 4 | The correlation structure between expansion patterns associated with ecological and behavioural variables reveals a putative ordering of cortical adaptation to habitat complexity. a** We used two-sided Pearson correlation ($n = 21$, the number of pPCA dimensions used in the analysis) between the matrices of pairwise effect measures of each observed ecological and behavioural variable to highlight the ordered nature of the effect of habitat complexity on the differential expansion of cortical areas. Significant correlation are indicated by *$p < 0.05$, **$p < 0.01$, ***$p < 0.001$. Statistical information is provided in Supplementary Data 4a. **b** Computation of relative measurement correlation requires a metric for the explanatory variables of the individual groups. A measure for the relative habitat complexity of fossorial species (−0.278) is computed from a tentative ordering of habitat complexity and by fixing the two more complex habitats that contain "arboreal" (1) and "terrestrial" (0) species. This scaling is applied globally to all relative expansion maps and does thus not bias the analysis of relative expansion of specific cortical regions with respect to each other amongst habitats. **c** Analysis of

the effects of habitat on the surface area of functionally defined regions in the cerebral cortex revealed a putative ordering related to habitat complexity. Violin plots of the distribution of local expansion values in each region for each map associated with a specific habitat are shown together with median values and first and last quartile +/− 1.5 interquartile range (IQR), as well as 95% percentiles of the output of the repeated measurements correlation model of the same values. The dorsal attention areas for terrestrial and fossorial and the ventral attention regions for arboreal and terrestrial species were the only regions showing no significant (two-tailed Friedman test) effects in terms of relative expansion, while the limbic areas where the only ones to show significant relative expansion in association to fossoriality (r = 0.628, $p < $ 1e-6, 95% CI = [0.584, 0.671], FDR-corrected). All statistical information is available in Supplementary Data 4b). Repeated measurement correlation analysis (shown as predicted correlation for each functionally defined region together with 95% confidence intervals) corroborated this result by indicating non-significant effects only for the ventral attention network (Supplementary Data 4c).

In extant species, the range parameter is consistently higher in primates (0.567–0.998) than in rodents (0.273–0.545, Fig. 6c; Supplementary Fig. 6; Supplementary Data 6a), indicating an increase in surface area attributable to multi-modal association cortex. In terms of socio-ecological parameters, the joint analysis of rodents and primates is dominated by the inter-order differences and showed significant difference in range for arboreality (r(88) = 0.397, $q < 0.001$, Δ = 0.338, 95% CI [0.094, 0.342]), fossoriality (r(88) = 0.488, $q < 0.001$, Δ = −0.375, 95% CI [−0.449, −0.291]) and group size (r(88) = 0.368, $q < 0.001$, Δ = 0.157, 95% CI [0.054, 0.233]) as well as diurnality (r(88) = 0.62, $q < 0.001$, Δ = 0.392, 95% CI [0.170, 0.385]). The value of Δ is positive if the range value is higher for the present compared to the absent socio-ecological parameter. Thus, arboreality, group size and

diurnality are related with less segregation, while fossoriality coincides with higher segregation of modality specific areas. In rodents, analysis of the relationship between ecological variables and spatial statistics of modal integration revealed significant negative effect for fossorial habitat (r(21.881) = 0.599, $q = 0.011$, Δ = −0.052 95% CI [−0.083, −0.021]) indicating higher segregation, whereas in primates, we observed increased modal integration in more gregarious species (r(56) = 0.374, $q = 0.01$, Δ = 0.088, 95% CI [0.029, 0.146]) as well as in diurnal compared to nocturnal species (r(41.206) = 0.57, $q < 0.001$, Δ = 0.0096, 95% CI [0.053, 0.1460]).

Finally, we estimated the evolution of spatial dependence of modal specificity by fitting semivariogram models on the ancestral state reconstructions of the cortical surfaces in the deep ancestral

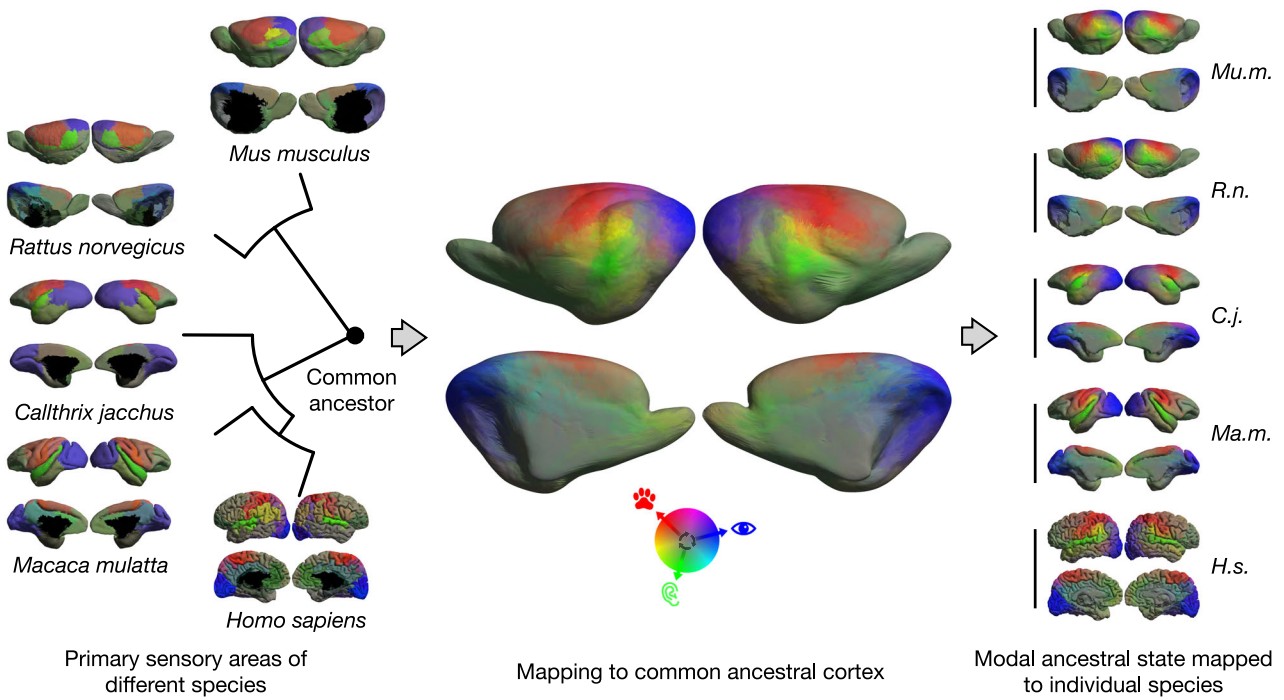

**Fig. 5 | The ancestral state of regional functional modal specialisation of cortical areas.** Cortical surface shape alignment across species and to common ancestors allows for the estimation of the spatial distribution of primary sensory areas (vision: blue, auditory: green, somatosensory: red) in the ancestral state of primates and rodents. The ancestral state can be mapped to all surfaces in the dataset to estimate the distribution and evolution of the relationship between modal specificity and surface shape. Renderings of the mapping of the ancestral state estimate onto all surfaces of the dataset are provided in Supplementary Data 11.

human lineage. We observed a strong positive relationship ($r(8) = 0.714$; $p = 0.002$; Fig. 6b; Supplementary Data 6b) between estimated divergence times and the spatial correlation range.

### Localised evolutionary cortical expansion in Homo

The fossil record does not allow for the precise reconstruction of the cortical shape, but much information about the behaviour and ecology of ancestral species can be gathered from skeletal and artifactual remains[53]. Mapping cortical surface models of all species to a common reference frame enables approximating the progression of local morphological changes in individual lineages during evolution. Provided the proposed alignment is correct, we expect the sequence of ancestral states of the cerebral cortex shape in the evolutionary history of humans to reflect adaptations to ecological niches and to therefore agree with established paleontological knowledge. At the same time, the reconstructed sequence enables a deeper investigation beyond the detail level of prior evidence.

In order to test this hypothesis, we calculated the expansion of brain regions between estimates of ancestral cortical shapes at branching points of the underlying deep phylogeny and decoded these maps by correlating them with human cortical activation patterns associated with neuroscientific terms[54]. At each time-point, we kept the 1% most strongly correlated terms, resulting in a description of the functional association to evolutionary changes in the cerebral cortex morphology (Fig. 7; Supplementary Data 7).

Hierarchical clustering (Supplementary Fig. 3) of the resulting expansion trajectories revealed 7 distinct evolutionary components

(silhouette coefficient 0.574) that summarise the functional correlates of the morphological evolution of the human cerebrum. The maxima of association between expansion and functional categories of the components revealed a sequence of neuromorphological stages, starting with the expansion of visual areas, followed by parahippocampal regions in the late Cretaceous, auditory and sensory-motor areas in the early Oligocene until the mid-Miocene and finally expansion of association areas in the late Miocene and Pliocene up until the present day.

We then compared the evolutionary expansion patterns in the two distinct lineages of mice and humans (Supplementary Fig. 4) as they represent the two most commonly studied species of Euarchontoglires. We calculated the relative area of each element in the cortical surface models in each lineage, interpolated these values at consistent timepoints and computed the pairwise correlation between each resulting vector. The resulting measure of progressive divergence between mice and humans during the evolution of their cortical shapes showed strong correlation with estimates of atmospheric oxygen levels[55] ($r(84) = 0.915$, $p < 0.001$, 95% CI [−0.944, −0.873], Supplementary Data 8a). That is, we observed accelerated divergence during periods of rapid decline in atmospheric oxygen levels. In order to quantify the origin of this divergence, we aggregated the areas of each model using a functional parcellation of the human cortex[46] (Supplementary Data 8b). This revealed both a significant increase in relative cortical surface area ($\beta(105) = 0.15$, $p < 1e-6$, 95% CI [0.1341, 0.1659]) as well as a positive interaction with ancestral time ($\beta(105) = 0.0017$, $p < 1e-6$, 95% CI [0.0014, 0.0020]) for the limbic areas of the cerebral cortex in mice

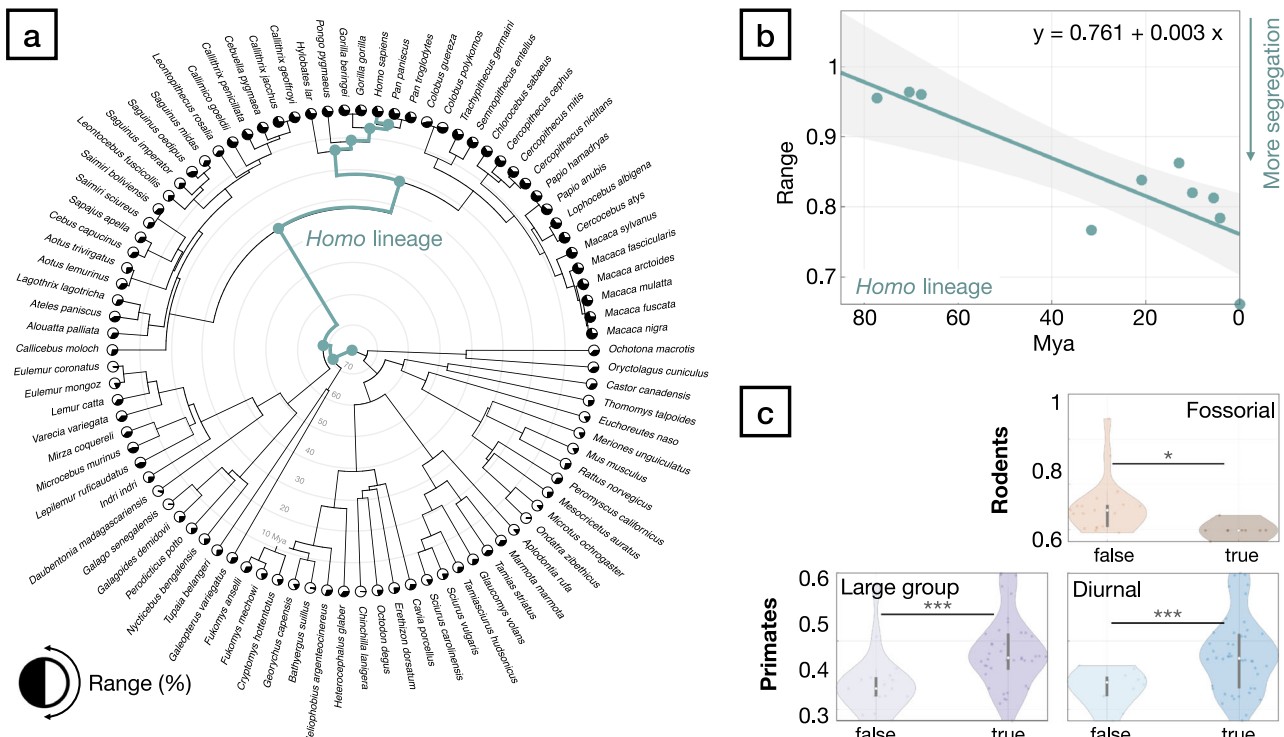

**Fig. 6 | Segregation of primary sensory cortical areas during evolution.** Mapping of species-specific cytoarchitectonic atlases allows for the determination of the functional topography of cortical regions and the relationship of the relative positioning of functional cortical areas and species ecology. **a** Spatial statistics indicate an evolutionary segregation of primary sensory cortical areas in the deep ancestral human lineage (green path), with a (**b**) significant positive relationship between the integration of sensory areas and evolutionary time (r(8) = 0.714, p = 0.002, shaded area indicates the 95% confidence interval of the linear model. See also Supplementary Data 6b). **c** Violin plots of the values for range parameters of the spatial statistics model for fossorial and non-fossorial rodents as well as

social/non-social and diurnal/nocturnal primates, together with median values as well as upper and lower quartile +/− 1.5 interquartile range. Significance of group differences of the range parameter for primates (n = 58) and rodents (n = 28) was assessed using two-tailed t-tests and FDR-corrected for multiple comparisons. Results show that in primates, group size and diurnality are associated (q < 0.001, see Supplementary Data 6a) with a higher range parameter, indicating more widespread spatial correlation of measurements of areal sensory specialisation. In contrast, in rodents, fossoriality is associated with a lower range parameter (q < 0.05, Supplementary Data 6a), indicating more segregated sensory processing areas in a subterranean habitat.

relative to humans, whereas the human lineage showed an increase of motor (β(105) = 0.0011, p < 1e-6, 95% CI [0.0007, 0.0014]) but also to a lesser degree default mode (β(105) = 4.3230e-04, p = 0.011, 95% CI (0.0001, 0.0008)) and frontoparietal (β(105) = 4.7530e-04, p = 0.005, 95% CI (0.0001, 0.0008)) areas with time, indicating a possible evolutionary sensory specialisation for olfaction in mice and for complex motor and association capabilities in humans.

## Discussion
In this study, we demonstrated how ecological and behavioural adaptations during evolution are mirrored in the shape of the cerebral cortex across extant species. Our analysis is based on a novel common geometrical reference frame for the cortical geometry of 90 species of Euarchontoglires. Using this common reference frame, we demonstrated that cortical morphology contains information on the evolutionary processes that produced its natural present diversity. The phylogeny of cortical shapes can be decomposed into individual components in both space and deep evolutionary time, both demonstrating concise functional interpretations and relationships to ecology. The spatial heterogeneity and inter-species variability of these processes highlights the importance of considering the evolutionary origins of the structures and functions studied in neuroscience, and the resulting inherent intricacies in translating experimental results between species.

By analysing the principal modes of variation of the dataset using activation maps obtained from human functional neuroimaging studies, we showed how neuroanatomical correlates of specific socio-

ecological niches are associated with the shape of the cerebral cortex in extant nonhuman species. Furthermore, we showed how the complexity of both their ecological and social environments is associated with the selective expansion of functionally specific cortical regions. Using spatial statistics, we demonstrated a gradual evolutionary decrease in the spatial range at which the modal specificity of these cortical areas becomes independent, suggesting the emergence of areas with multiple co-located associations[51].

By decoding the neurological function of the estimated cortical expansion in the deep ancestral human lineage, we were able to propose, for the first time, a sequence of both temporally and spatially circumscribed neuromorphological events that shaped the cerebral cortex of our ancestors over the past 77 million of years. Many of these events align with established results from paleoneurology such as an increase in the visual cortex in both primates and rodents[56–58] as well as the impact of arboreality[48,59,60] and fossoriality on the complexity of the cortex[48,60,61].

### Evolutionary neuroecology of group size and diurnality
In performing meta-analytic decoding of the cortical expansion associated with large social group size, we recover different aspects of what has been termed the social interaction network[62] (Fig. 3). We find that group size correlates with the expansion of anatomical regions of the prefrontal default mode network such as *cuneus* and *precuneus*, *cingulate*, *medial prefrontal cortex (mpfc)*, *posterior* and *anterior cingulate cortex (PCC, ACC)* that are known to be related to social behaviour[63,64] as well as areas associated with functional terms such *psychosis* and

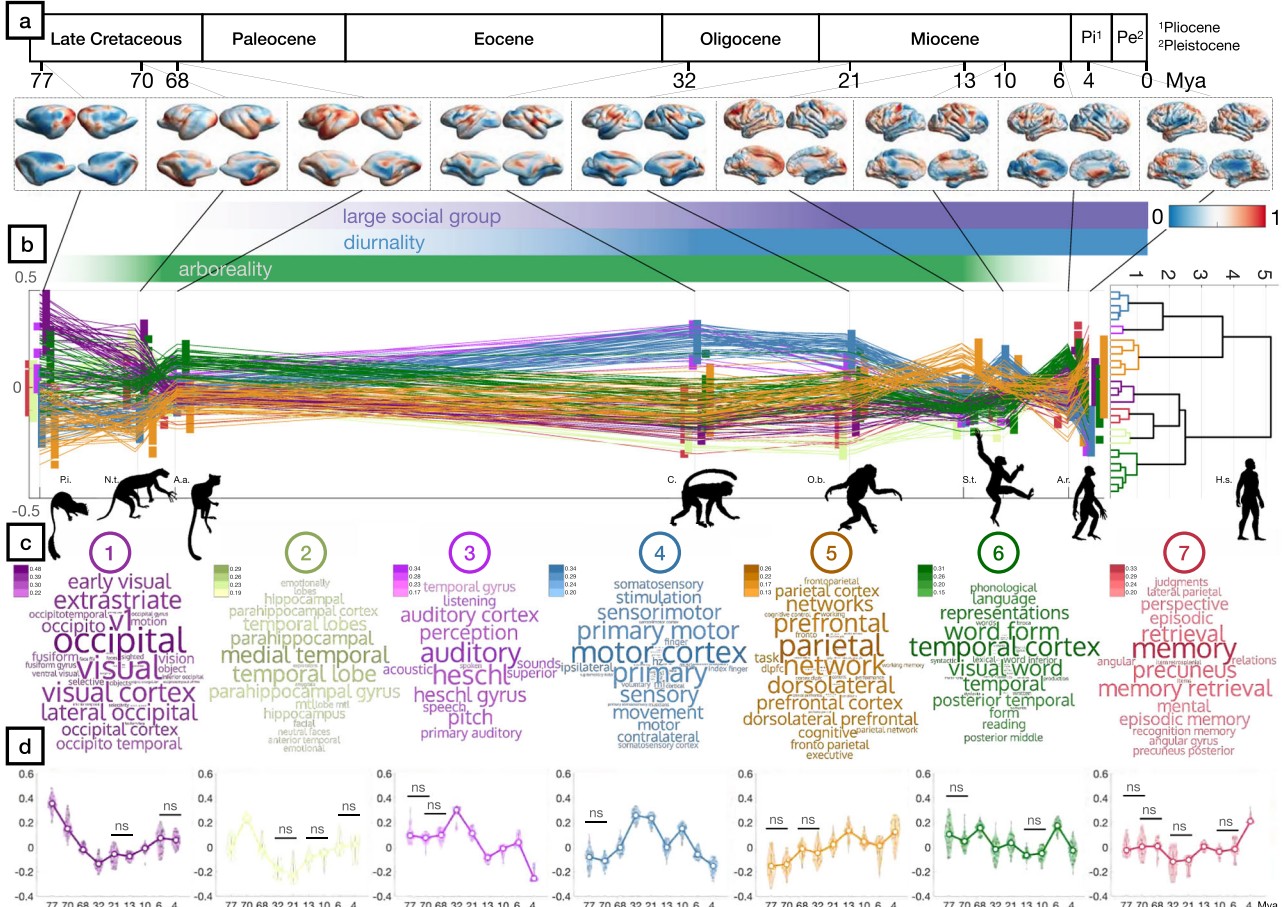

**Fig. 7 | Estimated sequence of local cortical expansion from the last common ancestor of rodents and primates to _Homo_. a** Consecutive expansion maps between the cortical surfaces are obtained from ancestral state reconstruction of all ancestors of _Homo_ until the LCA with Glires. **b** Decoding these expansion maps as correlations with statistical maps of human cortical function[54] and retaining the 1% most highest correlated maps at each time-point allows to perform (**c**) hierarchical clustering of the sequential correlation of expansion patterns in the deep ancestral human lineage (Supplementary Data 7a). **d** This reveals a decomposition of evolutionary cortical surface expansion that can be approximated as a sequence of steps (Supplementary Data 7b), starting with the expansion of primary visual, auditory and motor regions in the Cretaceous up until the Oligocene, followed by the expansion of higher order association areas in the Miocene, Pliocene and

Pleistocene. The common _x_-axis of all subplots (**a–c**) denotes million years ago (Mya). The colormap on the cortical surfaces (**a**) indicates relative area expansion between estimated speciation points. Markers for correlations in subplot (**b**) are jittered in horizontal direction to reduce overlap. Correlations are summarised for each cluster using violin plots in subplot (**d**) and joined at their median values through time. Statistical significance for differences between distributions was assessed using repeated measurements ANOVA. Statistical information as well as sample numbers for each cluster and time-point are omitted for space reasons and can be found in Supplementary Data 7c, license information for artwork used in Supplementary Data 7d. Abbreviations: P.I. _Ptilocercus iowii_. N.t. _Notharctus tenebrosus_. A.a _Archicebus achilles_, C. _Cercopithecoidea_, O.b. _Oreopithecus bambolii_, S.t. _Sahelanthropus tchadensis_, A.r. _Ardipithecus ramidus_, H.s. _Homo sapiens_.

_emotional responses_. Additionally, we observe a strong correlation with terms related to visual processing (_lingual gyrus, primary visual_) as well as the default mode network per se (_default mode, deactivation_).

While the ACC is considered integral in understanding the intentions of others and in observational learning[65], both PCC and dmPFC are crucial elements of the default mode network[66]. The PCC has been related to self-referential functions[67], and the dmPFC is deeply involved in processing observed social interactions[68] and judgements[69]. Together with the cuneus and precuneus, these regions have been proposed to form a "social-affective default network"[70] with strong links to face processing[71].

Expansion maps derived from ancestral state reconstruction of the cortical surfaces locate the time point of maximum correlation between cortical expansion and terms associated with large group size (ACC, cingulate, cingulate cortex) at the divergence between Homininae and Ponginae, some 12Mya. Compared to other great apes, orangutans (_Pongo_) are more arboreal and less social[72]. These results are indicative of an indirect influence of decreasing arboreality in african apes on brain evolution. Possible explanations for this

relationship lie in evolutionary advantages in predator avoidance posed by congregating in larger groups in a more terrestrial habitat[73]. While more arboreal apes can rely on the safety of their habitat, more terrestrial species have to more actively secure their environment by forming large social groups[74].

The association between social group size and activity pattern in primates is reflected in the correlation between variation in cortical morphology attributable to social group size and expansion of visual areas. Partial correlation analysis of the relationship of functional categories derived from cortical expansion with the evolution of sociality, controlling for diurnality (Supplementary Figs. 7 and 8; Supplementary Data 9 and 10a) revealed a significant positive correlation with area size associated with the terms _amygdala, hippocampus_ and _neutral faces_, and significant negative correlations with area size associated with terms associated with motor function such as _motor, premotor, finger, force_ (Supplementary Data 10b), highlighting elements of emotional control[75,76] as differentiating between effects of social behaviour and diurnality. While diurnality facilitates foraging for high-quality foodstuff, an actual increase in a species' dietary efficiency

requires the integration of more complex cognitive and social capabilities[77]. Contrary to social group size, we could find no significant relationship between activity pattern and cortical morphology. However, analysis of the variability of the functional topography of the neocortex revealed a stronger multi-modal integration in both group-living as well as diurnal primates, indicating a potential evolutionary strategy for success in cognitively more demanding environments.

Our results indicate cortical correlates of a strong evolutionary interplay[78] between this complexification of visual processing, social cognition[79] and ecology implicating the default mode network[80,81]. The ability of discriminating red hues might[82] be a crucial component in diurnal predator avoidance[83] and potentially foraging as well as for the interpretation of facial features[84]. These adaptations might be at the root of a positive feedback loop in encephalization whereby modifications to the functional topography of the cerebral cortex resulting from evolutionary changes to cortical morphology allowed for the integration of multimodal cues from complex ecological and social environments. This potentially increased our ancestors' capabilities for social behaviour and thus predator avoidance and effective foraging. This in turn led to increases in energy[85] available for supporting expensive brain tissue[86], enabling ever more complex social and cultural faculties[87,88].

### Variability of cortical morphology and topography in relationship to habitat complexity

Our findings revealed distinct patterns of cortical expansion associated with individual habitats (Fig. 3; Supplementary Figs. 5b–d, 6a, c). Arboreal species rely mostly on vision for perceiving their environment[89], which enables them to locate important objects such as food sources or predators at a distance and has, alongside the evolution of binocular colour vision and consequent depth perception, led to the complexification of cortical visual processing. This is reflected in our data by a selective expansion of occipital and dorsolateral aspects of the cerebral cortex in association with arboreality. Expansion of these regions of the dorsal pathway[90] likely reflect the fact that the functions it supports such as processing of object location, motion and concordant planning of movements are more relevant and also more elaborated in the complex 3D environment of the trees compared to the less intricate settings in which terrestrial and fossorial species live[91].

In terrestrial and fossorial environments, proprioception is algorithmically less challenging and perception at distance is, especially for small animals such as rodents, of lesser importance, as it is inherently limited by grass cover or subterraneity. Concurrently, we observed a graded reduction of the relative extent of visual areas in association with both terrestriality and fossoriality, related to the reduced relevance of visual information in these habitats (Supplementary Figs. 5c–d, 6c). On the other hand, terrestrial and fossorial species make stronger use of olfactory and gustatory cues for perception. This is reflected in the selective expansion of cortical areas associated with the processing of these chemical signals, in this case particularly the insular cortex. We observed rather fine-grained differences in the expansion of the insula in association with terrestriality and fossoriality, the former correlated with higher extent of its posterior, the latter with higher extent of its anterior aspect (Fig. 3). Terrestriality is additionally associated with selective expansion of the cingulate cortex. Meta-analytic decoding maps this pattern onto terms such as *nociceptive*, *noxius*, or *pain*. The cingulate is deeply involved in fear response[92] and considered as a mediator of emotional states and action selection[93]. This function might have increased relevance in terrestrial animals that rely on olfaction for survival by channelling otherwise functionally pervasive stress responses[94].

A fossorial lifestyle poses physical constraints on cranial shape[95] and limits the distance at which animals are able to perceive their environment, and burrowing brings the animals' gustatory system into closer proximity with chemical cues. The association of fossoriality with an increased expansion of the gustatory areas of the anterior insula (Fig. 3), as well as a relative expansion of the limbic regions of the functional parcellation of the cortex we observed (Supplementary Fig. 5d) can be explained by an adaptation to this environment. Furthermore, fossorial rodents also exhibited less modal integration than their arboreal and terrestrial counterparts (Fig. 6c), which might also relate to the fact that perception and interpretation of a comparatively impoverished ecological environment does not require the integration of multiple sensory modalities.

In contrast to the processes of mapping the position and motion of objects within the extrapersonal space, determination of object identity, as enabled by the ventral pathway, is less dependent on one individual sensory modality[96]. Studies in congenital blindness have repeatedly demonstrated the recruitment of these areas of the cortex for haptic processing[96–99], underlining the integrative role of the structures of ventral pathway[100,101]. This modal agnosticism is also reflected in the independence of the extent of the ventral attention areas on habitat observed in this study (Fig. 4c), which underlines the cross-modal relevance of the ventral processing stream for cognition.

An alternative or possibly complementary explanation for a relationship between the expansion of areas of the dorsolateral cortex and complexity of a species' environment can be found in the affordance competition theory of cortical organisation[102]. Under this framework, a selective expansion of areas of the dorsal stream in complex environments could lead to an increased complexity of proposed potential modes of action such as action maps[103]. Since the complexity of selecting amongst them however stays comparatively constant, no expansion of the ventral stream is required.

The evolutionary adaptations in the deep ancestral human lineage represent only one of many courses of cortical evolution. A common reference frame for the cortical shape evolution of different lineages enables the analysis of the rate of their divergence and its association with ecological and environmental factors. This analysis (Supplementary Fig. 4) shows that periods of rapid oxygen level[104,105] decline coincide with accelerated divergence between the cortical morphology in human and mice lineages, suggesting different adaptation strategies being fostered by the corresponding evolutionary pressure.

### Regional cortical expansion reflects sensory segregation and behavioural evolution

Hypotheses about the evolutionary history of primates are manifold[106] but have remained elusive to quantitative evaluation[22]. Our analysis of the joint morphological space of the neocortex of Euarchontoglires enables for the first time to study both the conserved topographical structure of cortical function as well as deriving a data-driven modular decomposition of its evolution. Our resulting estimate of the sequence of evolutionary steps of cortical development in the deep ancestral human lineage can be interpreted within the frameworks of neuroethology[107], ecomorphology[106], and phylogenetic refinement[15].

The gradually decreasing range of functional specificity on the cortex during the evolution of the deep ancestral human lineage (Fig. 6b) indicates a relative reduction of the size of primary sensory areas compared to emerging larger intermediate regions without unimodal assignment. In support of the tethering hypothesis[51], our results show that the modular segregation of primary cortical areas increased during human evolution. At the same time, the observed stronger modal integration appears to be favourable or even necessary in cognitively demanding tasks such as social interactions.

By splitting local cortical evolutionary expansion into clusters related to vision (Fig. 7, cluster 1), orientation (Fig. 7, cluster 2), temporal language areas (Fig. 7, cluster 6) and memory and retrieval (Fig. 7, cluster 7) on the one hand and auditory perception (Fig. 7, cluster 3), sensorimotor (Fig. 7, cluster 4) and prefrontal/dorsolateral/parietal

(Fig. 7, cluster 5) areas on the other hand, the observed topmost hierarchy of evolutionary functional expansion patterns reflects the dual stream hypothesis of cortical topography[90,108]. At the lowest hierarchical level in the ventral super-cluster, hierarchical clustering groups together patterns associated to parahippocampal areas (Fig. 7, cluster 2), memory and retrieval (Fig. 7, cluster 7). In the dorsal supercluster (Fig. 7, clusters 3-5), the grouping of expansion patterns related to auditory processing (Fig. 7, cluster 3), and sensorimotor function (Fig. 7, cluster 4) at the lowest level reflects the close relationship of these functions concerning proprioception.

From a historic perspective, our results indicate that the divergence of rodents and primates started with the development of visual processing areas (Fig. 7, cluster 1), likely adaptive to the increased sensory complexity posed by living in the complex 3D arboreal environment (Supplementary Fig. 5) as well as an increased visual information content due to binocular and chromatic vision[82]. This shift away from a previous ground-dwelling lifestyle led to an increase in foraging range in early primates[107], where the capability of forming and storing mental images of scenes such as locations of temporally limited food sources proved evolutionary advantageous, leading to the expansion of parahippocampal areas[109] (Fig. 7, cluster 2) specialised for spatial navigation and memory[110-113]. The increase in mobility resulted in the necessity of communicating over larger distances and thus the development of cortical areas related to vocalisation[114] and auditory processing[115-117], (Fig. 7, cluster 3). These evolutionary adaptations to arboreal locomotion and vocalisation occurred concordantly to the adoption of increasingly diurnal activity pattern and social behaviour (Supplementary Figure 5) and coincide with the development of complex motoric capabilities[103], which is reflected in the ongoing expansion of somato-motor areas (Fig. 7, cluster 4).

Due to the development of modality-specific processing areas, the primate brain around 21Mya may have been capable of performing complex visual, auditory and somato-motor signal processing, increasing the necessity for mechanisms for both selective attention and action selection[118] to maintain the ability to act. This may have been achieved by the expansion of prefrontal and parietal control areas in the cortex[17,119-121], (Fig. 7, cluster 5). The resulting capacity of selectively attending to specific stimuli entails the formation of complex mental representations[122,123]. Importantly, previously established executive control over complex motor systems enables the communication of these internal concepts in signed[124] or vocalised form[125], which has important evolutionary advantages[126] and leads to a complexification of social behaviour via the establishment of primordial forms of language[125,127,128].

Based on our results, these developments were accompanied by a shift back to a terrestrial lifestyle starting some 10Mya and ultimately bipedalism. This in turn potentially led to a freeing of the upper limbs from requirements of locomotion, and indirectly allowed for the development of greater fine muscle control[129] (Fig. 7, cluster 4), thereby enabling tool building, primarily for hunting[130]. The resulting shifts in nutritional efficiency would have reduced the selection pressure for enlarged masticatory apparatus and led to craniofacial changes which, together with the evolution of an orthograde body plan and the consequent rostral shift of the foramen magnum[53], would have allowed for the expansion of medial and lateral parietal regions of the cerebrum[131,132] (Fig. 7, cluster 7). These de-novo available cortical territories assumed auto-analytic functions of memory retrieval and episodic memory[133] which form fundamental faculties for establishing shared semantic concepts[134]. This establishment of social[135] and cultural capabilities could have created a positive feedback loop for increased encephalization[136] whereby capacities of Theory of Mind, abstract thought and the capability of externalising and transmitting this knowledge via language[137,138] and artefacts[124,139] may have been amplified with each generation of hominids, increasing the species' adaptive capabilities

and thus evolutionary fitness in increasingly harsh environments[140], giving rise to the *Homo sapiens* of today.

## Limitations

There remain numerous limitations to our work. For one, the breadth of our sample requires us to rely on single and oftentimes post-mortem exemplars of each studied species showing varying degrees of degradation. This, together with the inherent geometric limitations in mapping and averaging protruding features, might lead to an underestimation of the ancestral size of the olfactory bulb[41,46]. Furthermore, excessive evolutionary expansion of the neocortex in all extant species of Euarchontoglires may lead to an overestimation of its size[38,47], a bias which is especially present in the reconstruction of Cretaceous ancestors[48,49]. We also suspect that these data-dependent limitations, together with a range of unobserved covariates as well as intra-species variability lies at the root of the comparatively low part of extant morphological cortical variability being explained by the factors studied in this paper. Notably, the ecological categories used in the analyses represent stark simplifications of complex behaviours that encompass specific modes of locomotion, perception and social interactions. We don't propose that the results reflect a complete generative model of brain shape evolution. However, the recovery of significant influence of socio-ecological factors on cortical shape, the qualitative compatibility of its ancestral reconstructions with fossil endocasts and results obtained from quantitative analysis of its evolutionary development serve as evidence for the validity of the proposed common reference frame and the estimates of common ancestor cortices. We hope to see this tool being used and refined by other researchers using different modalities and ecological variables.

We also acknowledge the inherent anthropocentrism in the methods used for interpreting our results. Therefore, our description of the neuroanatomical foundations of human perceptive, interpretive and communicative capabilities can only serve as a schematic representation of processes that occur at multiple spatial and temporal scales, from genetic regulation to social interactions, from axonal depolarisation to evolutionary adaptation. Despite these limitations, to our knowledge the data presented in this paper represents the most diverse collection of Euarchontoglires brains analysed to date and provides important insights into the origins and ramifications of cortical functional topography.

## Methods

We performed geometric cortical shape alignment across species, and subsequently modelled the phylogeny of the cortical differences among extant species. First, we extracted a mesh surface of each cerebral hemisphere from available segmentations (Fig. 2a). Labels for anatomical structures were projected onto the surface by volumetric interpolation and converted to pseudo-probabilities using a gaussian kernel on the geodesic distances on the mesh. Surface features describing local mesh properties were computed from the surface models. In parallel, the global geometry was encoded via the spectral embedding of the Laplacian Matrix associated with the surface mesh. These two sources - anatomy and geometry - of information were combined on a per-vertex basis to encode the local cortical shape. We established correspondence between the cortical surface meshes of two species, by matching their cortical shape encodings in a high-dimensional embedding space. Dense surface correspondences could then be established from the resulting alignment between these two aligned representations (Fig. 2b). To estimate the potential shape of the cortical surface of an ancestor of two sister species in the dataset, the ancestral state reconstruction was first performed on global shape properties. This allowed the positioning of the ancestral shape along the geodetic connection in the space of smooth shells between the global shape properties of the two species. Since linear interpolation does not reflect the geometric properties of the surfaces, shape

interpolation was performed in the space of smooth shells to estimate the ancestral shape (Fig. 2c). In order to iterate the procedure a-c and estimate further ancestors, the label information of the two sister species was mapped onto the estimated surface mesh of their common ancestor. If a label was present in both sister species, information was combined for estimating its ancestral state. However, similarly to the reconstruction of the ancestral surface model, computation in the Euclidean space did not account for the properties of the domain of the pseudo-probabilities and would have led to blurring and information loss. Instead, optimal transport on meshes was used to recover an estimate of the extent and shape of the ancestral label pseudo-probabilities (Fig. 2d). This procedure was then iterated by following the phylogenetic tree to its root.

A more detailed description of the technical aspects of the methods employed for phylogenetic modelling, surface reconstruction, surface matching and estimation of spatial statistics can be found in the Supplementary Methods.

### Dataset description

**Brain imaging.** For the present study, we collected imaging data of 90 different species from various sources (Supplementary Data 1a), consisting of 58 primates, 28 rodents, 2 lagomorphs, 1 scandentian and 1 dermopteran. In order to collect a maximum number of different species, we combined imaging data from different modalities of both in-vivo and ex-vivo specimens. In total, 75 specimens were imaged using MRI, 7 using diffusible iodine-based contrast-enhanced Computed Tomography (DiceCT) and 8 using serial histological sections. For the serial histology section data, StackReg[141] was used to correct for rigid displacement between slices. Data for 10 species was available in vivo, while 80 scans were obtained from ex vivo data. Of the 80 ex-vivo specimens, 43 were fixated ex-situ, while 37 were imaged inside the skull. The data for 7 species consist of atlas averages of multiple individuals. For 7 specimens, previously published segmentations of the cerebral structures were available. For 31 specimens, segmentation of the cerebrum was available via the BrainCatalogue[5]. For the remaining 52, we manually segmented the two cerebral hemispheres using ITK Snap[142]. In cases where it remained intact after preparation, the olfactory bulb was manually segmented as a separate label, as well as the Corpus Callosum. We further included commonly used geometric spaces for model animals (*Mus musculus*, *Rattus norvegicus*, *Macaca mulatta*, *Callithrix jacchus*, *Homo sapiens*) in the dataset (Supplementary Data 1a). Unfortunately, it was not possible to obtain multiple individual scans for all species. We therefore limit our dataset to one exemplary shape per species, as the morphological variability is limited for the cerebral cortex in most species. For humans, where anatomical variability is especially large, we use the "Colin" template space[143].

**Surface model construction.** We computed surface models individually for each cerebral hemisphere in each specimen. In the case of 7 species (Supplementary Data 1a) where either only one hemisphere was available or one showed defects that were too large for manual correction, the model of the contralateral one was mirrored along the midline to replace the missing hemisphere. Spherical topology of the resulting surface model was ensured by respective functionality provided by the CAT toolbox[144] as well as FreeSurfer[43]. Spherical mappings of the resulting surface models were constructed[145] to serve as parametric reference frames for geometry processing.

**Ecological and behavioural data.** Ecological and behavioural data were collected from literature sources (Supplementary Data 2). We collected data for daytime activity pattern and social group size, which we simplified into binary categories of "Diurnal" and "Large social group". We also collected data for preferred habitat, which are more fragmented and harder to distil into viable categories. In order to retain statistical power while capturing the essence of these complexities, we abstracted species habitat in the following way: Arboreal - spends most of its time in trees, in a cluttered environment, is able to move in all directions; Terrestrial - spends most of its time on the ground, in an open-environment, does not habitually move in vertical direction; Fossorial-spends most of its time underground, affecting its range of movement and perception. Individual species can belong to multiple categories in cases where the literature is ambiguous. While this classification does not do justice to the complexities of ecological and behavioural diversity observed in the species under study, we found this nomenclature to be the most consistent among such a large diversity of rodent, lagomorph, primate, dermopteran and scandentian species. Species that were reported as either diurnal, cathemeral or arrhythmic in the literature were labelled as "diurnal". Species that are not reported to be solitary or pair-living were labelled as living in a "large social group".

**Cortical shape encoding.** We performed pairwise matching of the cortical surface models of sister species in the underlying phylogenetic tree using methods from computational geometry. Specifically, we exploit the fact that the eigenvectors of the cotangent graph laplacian[146] of a mesh representation of a shape are closely related for isometric shapes. This property has been leveraged in numerous geometric applications, most notably shape description[147],matching[148] and alignment[149].

As we are however specifically interested in deviations from isometric scaling due to localised expansion of individual cortical areas, direct application of spectral matching would lead to erroneous alignment. Therefore, we augment the global intrinsic spectral representation of the shapes by additional extrinsic features representative of cortical morphology. Since the patterns of cortical folding show strong evolutionary conservation, we use sulcal depth maps, their surface derivatives, gyrification indices as well as the Hamilton–Jacobi Skeleton of surface curvatures[150] as descriptors of the sulcal geometry. Furthermore, we include manual labels for the olfactory bulb, the corpus callosum and the cerebral medial wall as scalar values defined on each surface vertex of each surface model. The label maps are generated by manually labelling the individual structures in the volume, interpolating these binary maps onto the surface using Nearest Neighbour interpolation and consecutively computing the geodesic distance transform[151] on the surface to obtain a continuous function value for each surface vertex. The geodesic distance is then subjected to an exponential kernel to localise the label information.

**Dense surface alignment.** From the spectral decomposition of the mesh Laplacian and the extrinsic surface features, we obtain a vector representation of global and local shape information at each surface vertex, which we consider elements of a combined spectral/euclidean embedding space. Matching of elements of that space is performed using Coherent Point Drift (CPD)[152]. CPD is a statistical formulation of the problem of shape correspondence based on Gaussian Mixture Models and solved using Expectation Maximisation. In this formulation, the feature vectors corresponding to the vertices of one shape represent the centers of Gaussian distributions in a high-dimensional space, whose positions are optimised such that the feature vectors of the other shape are realisations of the resulting mixture of Gaussians under Maximum Likelihood. After convergence, correspondences between points can simply be obtained by selecting the center of the Gaussian with the highest weight for each surface point. This probabilistic formulation relaxes the isometry assumption of the original spectral matching and simultaneously serves as regularisation for the ill-posed shape matching problem, effectively penalising strong local stretching and bending[149,153].

**Post-processing of surface alignment.** Due to the nonlinear correspondence between the original Euclidean three dimensional space in which the cortical surface models are observed and the spectral embedding space, additional processing of the correspondences is required in order to obtain continuous, invertible maps between two cortical surface models. First, we use a method proposed for the matching of developing fetal brain surfaces in humans[154], which shows a comparable range of geometric variability as our dataset from lissencephalic first trimester of gestation to strongly gyrated in the third trimester. Instead of defining corresponding points in the two shapes directly using nearest neighbours in the embedding space, this method first performs edge-based smoothing of the embedding coordinates, resulting in smoother maps. In order to further enforce diffeomorphic correspondences between the shapes, the resulting correspondences are used as inputs to a joint Laplacian matching step[149,155]. Contrary to the original procedure of computing alignments between individual embeddings of the two shapes, this method is based on the spectral decomposition of a joint Laplacian matrix. The joint Laplacian of two shapes has a block structure, with the mesh topology and geometry of the individual meshes encoded in two diagonal blocks, containing edges between neighbouring vertices weighted by their geodesic distance, and the pair-wise correspondences between the two shapes encoded on the off-diagonal blocks, either as binary entries for corresponding vertices or also weighted by their respective distances in their aligned embedding space. As before, correspondences of one mesh to vertices of the other can be established using nearest neighbours in the resulting embeddings space of the now joint Laplacian. The third regularisation step that we perform in order to ensure reversibility of the resulting maps is based on the Dirichlet energy of the resulting map. As of now, we only considered the locations of correspondences of the vertices of one surface mesh in the other. However in practice, it is important to ensure that both maps from either mesh onto the other are well-behaved, eg. are reversible with as little distortion as possible. To ensure this, we optimise the correspondences obtained from the joint Laplacian embedding using the method proposed in Ezuz et al.[156], which explicitly optimises a joint energy functional consisting of terms for smoothness and reversibility of the resulting deformations using block coordinate descent.

**Approximating ancestral shapes of cortical surfaces.** We iterate the procedure of establishing pairwise correspondences between the surface models of the cerebral cortex in sister species, traversing the phylogenetic tree from the leaves until the root, corresponding to the last common ancestor (LCA) of rodents and primates. Enabling this iterative procedure requires the interpolation between matched surfaces. However, simple linear interpolation via weighted averaging in Euclidean space can result in both geometric and topological errors such as excessive thinning of structures and fold-overs due to the intrinsic nonlinearity of shape space. We therefore perform shape averaging in the space of discrete shells[41,42]. This allows the incorporation of biomedically motivated penalties to the calculation of intermediate shapes between shapes of matched topology. The underlying biomechanical model is equivalent to a thin elastic shell, which is only a rough approximation of the non-linear material properties of the true cortical tissue[157]. However, a complete incorporation of the biomechanics of brain tissue in the matching procedure is both mathematically and computationally challenging due to the non-linear tissue properties, and of questionable validity due to the unknown evolutionary changes they might be subjected to. We therefore leave these extremely interesting questions for future research and limit the biomechanics of the matching procedure to linear elasticity.

We could obtain the relative weightings of the surface models of each sister species in the computation of their average directly from the underlying calibrated phylogeny. However, this would imply the unrealistic assumption of uniform relationship between genetic similarity on which the phylogeny is based and the morphological differences in the shapes of the cerebral cortices of sister species. We therefore first estimate the ancestral states of global shape variables in the whole tree which does not requires a-priori correspondences (cf. *"Phylogenetic Model of Global Cerebral Morphology"* for a detailed explanation). This allows for the computation of a parameter reflecting the estimated scaling factor between genetic and morphological differences between an ancestor and its daughter states. We then generate the shell space interpolation at these distances between the two sister species as an approximation of the ancestral state of the cortical surfaces.

**Label fusion using optimal transport.** After establishment of pairwise correspondences between sister species, available annotations of specific homologous structures (Corpus Callosum, olfactory bulb, thalamus, hypothalamus, septum) are mapped onto the ancestral morphology estimated via shell-space interpolation. In cases where the same label is available for both daughter shapes, this information needs to be combined in the ancestral state. As with the shapes themselves, label information can only be poorly processed using euclidean methods, as linear averaging would inadvertently result in blurring of the boundaries of the labels due to both misalignment of structures due to regularisation in the matching procedure but also changes in the true extent of individual structures during evolution. Instead, we represent labels as pseudo-probabilities of the extent of a structure at a specific position of the cerebral cortex and use optimal transport geometry to compute averages of these distributions on the estimated surface models of ancestral states using Convolutional Wasserstein Distances[158]. This procedure results in estimates of the distribution for each label in the ancestral state, but also breaks the commutativity of label correspondence between parent and daughter states. We therefore perform an additional matching step between the newly labelled ancestral shape and its daughter shapes to reestablish consistency between the shape and label correspondences.

**Consistent topology of cortical morphology.** After iterative pairwise matching of the surface models of individual cortical shapes, global correspondence between all shapes in the dataset can be established by propagating the correspondences throughout the phylogenetic tree. Throughout the iterative surface mapping, we use the more densely sampled cortical surfaces of the two sister shapes as topology of the estimated ancestral shape. Once the LCA of rodents and primates at the root of the tree is reached, we resample all shapes in the dataset to fsaverage6 topology to allow the easy application of published results on cortical topography in humans. The resulting surface models are available at https://github.com/cirmuw/EvolutionOfCorticalShape.

## Phylogenetic modelling
**Estimation of phylogenetic relationships.** We obtain the phylogenetic relationships for the taxa used in the present study from a time-calibrated phylogeny of the mammalian class[39]. The phylogeny is constructed using a Bayesian "backbone-and-patch" method from a 31 gene Supermatrix, resulting in probabilistic estimates of the branch lengths and divergence times in the tree. We draw 100 samples from the posterior distribution of trees of the 90 species contained in our dataset and compute the consensus tree to obtain a single phylogenetic topology and divergence times. As the human brain showed extensive expansion compared to other species, we augment the resulting consensus tree with that of hominin fossils obtained from Melchionna et al.[159]. We merge the two trees using phylogenetic inference under matrix representation[160].

**Phylogenetic modelling of global cerebral morphology.** Estimation of the ancestral shape of the cortical surface models of two sister species requires a measure of the degree of morphological changes each sister species has been subjected to. However, divergence time estimates in the used phylogeny are based on genetic information, and there is no unique universal scaling factor that would relate changes to the genetic code to cortical morphology. On the other hand, estimation of an evolutionary model for changes in cortical shape requires a priori correspondences between surfaces. To avoid this circular problem, we assume a linear relationship between the changes to the global shape measures of surface area (SA) and gyrification and the degree of local changes to cortical morphology. As a first step in our phylogenetic analysis, we then use the combined tree of extant and extinct species to determine an appropriate evolutionary model for these global morphological properties of the cerebral cortex.

From the available 90 species, we estimate both the cortical SA and the area of the convex hull of each hemisphere, as their ratio constitutes a measure of gyrification[161]. We then fit scalar Ornstein-Uhlenbeck (scOU) models with evolutionary rate shifts[40] to the SA measurements of the cerebral cortex and its convex hull. Briefly, the scalar formulation of the OU mixture models simplifies the definition of rate shifts in multivariate correlated traits in a phylogeny by shifting the inter-trait covariance to the Brownian motion part of the OU model while modelling the selection pressure jointly as a scalar parameter for all observed traits. We assume that this simplification is acceptable for modelling the evolution of the SA of the cerebral cortex and its convex hull due to the existence of strong scaling laws[162–164] relating the folding and scaling of mammalian cerebra. Estimation of rate shifts is then performed in an Expectation-Maximisation framework, which allows for the automatic determination of evolutionary rate shifts. However, the scalar formulation of mixture of OU models is defined only for ultrametric trees (eg. trees whose tips all have the same age). We therefore shift the tip ages for the fossil data to the present day. For the extinct hominin species included in the augmented tree, only estimates of the SA of the convex hull of the brain are available from endocasts of the fossil crania; the SA of the cerebral cortex is considered missing in phylogenetic modelling. We assume that the slight misspecification of the tip-dates for the fossil hominin data is immaterial to the relative BIC values used for model selection. We consequently perform ancestral state estimation of the two SA values for each case and hemisphere and prune the tree to cover only the available extant species for further processing. The resulting estimates of the surface areas of the cerebral cortex and its convex hull are then available throughout the phylogenetic tree. We use the ancestral state reconstructions of these values to determine the interpolation factors between two matched cortical surface models to reconstruct their putative ancestral state without a priori matching of all shapes in the dataset as described above.

**Phylogenetic geometric morphometrics of cortical morphology.** After having obtained an estimate for the rate of evolutionary change in global morphological properties of the primate and rodent cerebrum, we aim at determining the shape space of the cerebral cortices of rodentia, lagomorphs, scandentia, dermoptera and primata. However, the dimensionality (40962 vertices per hemisphere, as defined in FreeSurfer fsaverage6 space) of the dataset is too large in relation to the sample size (90 cortical surface models). We therefore use low-resolution icosahedral subdivision to first reduce the dimensionality of the data to 162 vertices per hemisphere, and remove any samples in the medial wall, which results in 900-dimensional representations of each pair of hemispheres for each taxon in the dataset. We fit both simple (Brownian Motion (BM), Early Burst (EB) and Ornstein-Uhlenbeck (OU)) as well as OU models with evolutionary shifts (OU_shift) to the shape data[165]. Note that the position of these evolutionary shifts have been determined a priori from the global cortical shape parameters,

and the multivariate OU models used for modelling the localised evolution of the cortical shapes are not subject to the same simplifying assumption as those used in estimating the rate shifts. This approach results in a strong link between changes in evolutionary rates of the global and local cortical morphology, but nonetheless allows for the independent estimation of correlation structure for the local change in cortical morphology during evolution.

We perform phylogenetic model selection based on the Generalized Information Criterion (GIC)[166], which selects an adaptive OU model with 2 rate shifts (BM: -460733.9, OU: -480972.5, EB: -460731.9 OU_shift: -496364.8) (Supplementary Fig. 1). We perform phylogenetic PCA[165] on the low-resolution representation of the dataset in order to account for phylogenetic correlation in the shapes. We then use L2-regularisation as proposed in Clavel et al.[167] to estimate the phylogenetic covariance matrix, as its unregularised computation is still ill-posed due to the large number of landmarks compared to the number of samples. The ratio of one to ten between dimensionality of the samples and their number is comparable to that used in other works on phylogenetic brain shape analysis with similar or smaller sample sizes[168].

## Analysis methods for cortical surface maps

**Meta-analytic decoding.** We use meta-analytic decoding to analyse the cortical surface maps obtained from phylogenetic modelling (pPCA, ancestral state reconstruction). We first map the measurements obtained on the cortical surface into the standard MNI space[46]. The resulting volumetric maps are then correlated with 3228 individual maps obtained from 14371 individual neuroscientific studies of brain activation in humans[54]. We retain the most strongly correlated 1% of these terms to interpret the expansion maps in terms of neuroanatomical localisation and neurological function (Supplementary Datas 3 and 4). We note the anthropocentrism inherent to the utilisation of human reference maps for the decoding maps related to evolutionary processes. However, to the best of our knowledge there unfortunately does not exist to date any comparable dataset for other species which would improve on this.

**Projecting maps onto functionally defined parcellation.** Additionally to interpreting the maps obtained from phylogenetic modelling using meta-analytic encoding, we also analyse the obtained values in relation to their position in a commonly used parcellation of the human cortex from resting state functional MRI data[46]. While this sort of analysis can be criticised for its anthropocentrism, the proposed parcellation has been shown to be highly consistent for the analysis of a detailed battery of cognitive tasks[169], highlighting the robustness of the underlying topography in describing functionally distinguishable regions of the cerebral cortex.

**Merging of cytoarchitectonic and functional atlases.** Cytoarchitectonic cortical atlases are the maps that have been the most consistently defined for different species[170–172]. Specifically, detailed atlases have been published for *Mus musculus*[173], *Rattus norvegicus*[173], *Callithrix jacchus*[174], *Macaca mulatta*[175] and recently *Homo sapiens*[176]. Additionally, continuous work in analysing functional specificity of individual cortical regions has led to the proposition of a detailed atlas of the specificity of cortical areas in terms of auditory, visual and motor processing[52] in humans. We combine these two sources of information about the structure and function of cortical regions in order to analyse the changes their organisation was subjected to during evolution. Specifically, we first map the values for modal specificity into the individual highest resolution cytoarchitectonic parcellations in each of the 5 species for which cytoarchitectonic atlases are available, yielding 3 maps ranging from 0 to 1 at each vertex for each species. We estimate the ancestral state of each of these maps using the phylogenetic covariance matrix obtained from the cortical surface shape models. This

procedure can be thought of as a constrained surface smoothing procedure in which the blurring of functional specificity of individual regions in the cortex is constrained both by the species-specific cytoarchitecture and the phylogenetic stability of these maps.

### Statistical treatment

**General statistical methods.** Throughout the paper, significance was assumed at a 5% false positive level. When required, multiple comparison correction was performed using Benjamini-Hochberg correction for False Discovery Rate. Normality of distributions of analysed data was assessed using one-sample Kolmogorov-Smirnov (KS) tests. Dependent and independent *t*-tests were used for determining significance of group differences when normality could be assumed for both tested distributions. Wilcoxon rank sum tests were used In cases where normality could not be assumed in either of the distributions. When testing for equality of the means of multiple dependent measurements over time, we use repeated measurements analysis of variance (RANOVA) after testing for normality using KS tests at each timepoint, followed by post-hoc testing using the Games-Howell method. Equal variance was tested for using Bartlett, sphericity using Mauchly tests. In case of violation of the sphericity assumption, the Greenhouse-Geisser approximation was used for p-value adjustment. In case of unequal variances, the Games-Howell method was used for multiple comparison post-hoc testing.

In cases where independence of the measurements could not be assumed (see "Null models for brain maps" below), significance of group effects was established by permutation testing against effect distributions. Group effects were determined using Kruskall-Wallis tests, post-hoc testing using pairwise Wilcoxon-Rank-Sum/Mann-Whitney *U*-Test that were corrected for multiple comparisons using the Benjamini-Yekutieli procedure for non-independent tests[177].

**Testing for the effect of species habitat on functionally defined brain regions.** We use repeated measurements correlation (RMCORR)[47] for testing the association of non-independent measurements in brain regions with species habitat, where we consider an ordering of habitat in terms of sensory complexity from arboreal to terrestrial to fossorial. To enable such an analysis, we first compute the overall brain expansion computed from linear combinations of all pPCA modes significantly associated with a specific habitat. We use summary measures for each habitat to transform the ordering of habitats into a tentative continuous variable (Fig. 4a). The correlation values of each functional cortical area from RMCORR then correspond to a measure of relative consistency in the expansion or contraction of a specific region relative to the whole cortex.

**Null models for brain maps.** Measures obtained from different locations on the cortex violate the independence assumption, which precludes the application of most parametric and non-parametric procedures for estimation of significance of results. When needed, we therefore rely on permutation testing under appropriate null hypothesis. Specifically, when estimating the significance of results corresponding to a specific subregion of a brain map against others, we use a test distribution of 1000 randomly generated brain maps showing the same spatial autocorrelation at different locations of the brain obtained from a generative model[178]. The corresponding null hypothesis is that the observed effect is related to spatial autocorrelation of the measurements and not any process-specific topographic feature[179].

**Spatial statistics.** We use spatial statistics to quantify the relationship between distances on the cortical surface and differences in the distribution of cognitive modalities at these vertices. We approximate the geodesic distance between two surface vertices using the shortest path in the graph defined from the surface mesh topology, normalise the resulting value to the range [0,1] and bin the resulting values into 20 non-overlapping bins. For each bin, we compute the average pairwise cosine distance of the three-dimensional (propensity for visual, auditory and somatosensory function) vectors measured at the included surface vertices, which we again normalise to unit range. We then perform 100 independent, randomly initialised fits of linear, spherical and exponential models to the resulting empirical variogram and retain the model best-fitting model in terms of sums of squares error (SSE). From the selected model, we estimate the sill (maximum value of spatial independence) and range (distance at which the sill is reached, e.g. measurements of modal specificity at pairs of points separated from each other by a higher geodesic distance can be considered independent), where only the latter is of interest due to the a-priori unit-normalisation of both regressor and regressed variables.

**Software.** All analysis was implemented in Matlab R2014a and R2019a, Python 2.7.13 and 3.7.3 as well as R 3.6.0 and 4.0.3. Additional processing was performed using Convert 3D 1.1.0, FSL 6.0.4, FreeSurfer 6.0.0 and 7.1.1, ANTs 2.3.4 and ImageJ 1.49 u.

### Reporting summary

Further information on research design is available in the Nature Portfolio Reporting Summary linked to this article.

## Data availability

All quantitative data supporting the findings of this study are provided as Supplementary Information to the article. Sources of all imaging data used in the study as well as reference publications are available in Supplementary Data 1. Aligned surface models used to define the proposed common reference frame, as well as ancestral state estimates obtained from it are publicly available at https://github.com/cirmuw/EvolutionOfCorticalShape and https://doi.org/10.5281/zenodo.7713847. Expansion maps used for meta-analytic decoding are additionally made publicly available at https://neurovault.org/collections/IHFSXSES/.

## Code availability

Computer code written to perform surface alignment and analysis will be available from the corresponding author upon reasonable request.

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

## Acknowledgements

We gratefully acknowledge everyone who contributed data to the project. In alphabetical order, Eike Budinger of the Leibniz Institute for Neurobiology for contributing data of Meriones unguiculatus, Marc Dhenain of the Institut de biologie Francois Jacob for contributing data of Microcebus murinus, Hao Lei of Huashan Hospital for contributing data of Tupaia belangeri, Kyoko Miura of Kumamoto University for contributing data of Heterocephalus glaber, Thomas Morrison for contributing data of Mesocricetus auratus, Brian Tainor and Vanessa A. Minnie (with support from NIH R01 MH121829) for contributing data of Peromyscus californicus, Cody W. Thompson of the University of Michigan Museum of Zoology for contributing data of Tamias striatus and Sarael Alcauter Solorzano of the Universidad Nacional Autonoma de Mexico and Jason R. Yee of the University of Veterinary Medicine Vienna for contributing data of Microtus ochrogaster. We especially want to acknowledge the effort of making neuroanatomical data publicly available made by the Digital Morphology Museum of Kyoto University, the Brain Catalogue from Institut Pasteur, the National Natural History Museum Paris and the Institute for Brain and Spinal Cord, the NIN Primate Brain Bank of Utrecht University, the Institute of Neuroscience and Medicine, Brain and Behaviour (INM-7) at Forschungszentrum Jülich, the Comparative Mammalian Brain Collections of the Brain Museum from the University of Michigan, Michigan State University and the National Museum of Health and Medicine, the Montreal Neurological Institute and COBRA Laboratory at McGill University, the Laboratory of Brain and Cognition at the National Institute of Mental Health, the Laboratory for Symbolic Cognitive Development at RIKEN Brain Science Institute, the Center of Advanced European Studies and Research in Bonn, the National Institutes of Mental Health, the Duke Center for In Vivo Microscopy at Duke University, the Cambridge Biotomography Centre at Cambridge University, the Museum of Vertebrate Zoology at Berkeley, the Marseille Functional MRI Center, the Institut de biologie Francois Jacob, the Institute of Basic Medical Sciences at the University of Oslo and the Museum Nationale d'Histoire Naturelle de Paris. Furthermore, the authors gratefully acknowledge the collection efforts by the American Museum of Natural History (AMNH FM 12506, AMNH 4756, AMNH FM 128167, AMNH FAM 65511, AMNH 12561), the United States National Museum (Smithsonian Institution) (USNM V 17996, USNM 256584, USNM 17161), the Natural History Museum London (NHM PV M 1345), the Geological Museum of the University of Wyoming (UW 12362), the Field Museum of Natural History (FMNH UC 1642), the Musée d'Histoire Naturelle de Montauban, Victor Brun (MA PHQ 289), the Jackson School of Geosciences at the University of Texas, Austin (TMM 40688-7), the Yale Peabody Museum (YPM 14737) and the Museo Argentino de Ciencias Naturales, Buenos Aires (MACN 14128) as well as Doug Broyer, Mathew Colbert, Rachel Ives, Ruth O'Leary and Mary Silcox for providing access to endocast data. ES has received financial support for this project by the

Austrian Research Fund (FWF) grant I 3925-B27 in collaboration with the French National Research Agency (ANR). RT and KH were funded by the French National Research Agency and National Science Foundation grant agreement project NeuroWebLab (ANR-19-DATA-0025), DMOBE (ANR-21-CE45-0016), the European Union's Horizon 2020 research and innovation programme under the Marie Skłodowska-Curie grant agreement No101033485 (KH Individual Fellowship), and the STIC-AmSud programme project STIC-AMSUD + CLANN 22-STIC-03. GL has received funding by the Austrian Research Fund (FWF) grant P 35189, the Vienna Science and Technology Fund (WWTF) grant 10.47379/LS20065, and the European Commission grant 765148 TRABIT. OCB has received financial support from the Marie Skłodowska-Curie Actions: Individual Fellowship (H2020-MSCA-IF-2018-2020; No. 792611).

## Author contributions

E.S. designed the study and the experiments, created the software used in the analysis, performed the analysis and parts of the cortical segmentations and wrote the article. K.H.N. co-designed the study and the experiments, supported the analysis and co-wrote the article. K.H. initiated the project, performed parts of the cortical segmentations, interpreted the results and commented on and edited the article. N.J. performed scanning of specimens, interpreted the results and commented on and edited the article, O.C.B., provided fossil endocast data, interpreted the results and commented on and edited the article, R.T. initialised the project, interpreted the results and commented on and edited the article, G.K. provided funding for the project, interpreted the results and commented on and edited the article, D.P. provided resources for the project and edited the article. G.L. supervised the project, co-designed the study, interpreted the results and co-wrote the article.

## Competing interests

The authors declare no competing interests
