## [Peer Review File · Nature Communications]

Evolution of Cortical Geometry and its Link to Function,
Behaviour and EcologyReviewers' Comments:

Reviewer #1:

Remarks to the Author:

I should note at the start of this review that I'm not an expert in morphological/spatial statistics (so I cannot evaluate the validity of these aspects of the paper), though I am an expert in phylogenetic comparative methods and brain evolution.

In my view, this article presents the most convincing case to date that "ecological and behavioral adaptations during evolution are mirrored in the shape of the cerebral cortex across extant species". However, the balance of emphasis of the paper perhaps weighs too heavily on making this point, and too lightly on emphasizing the new insights these results may provide.

Overall, the results are intuitive and confirm what one would expect both for human brain evolution and primate brain evolution compared with rodent brain evolution. The clarity of the results are impressive, but I'm not wow-ed by landmark new findings (aside from demonstrating the validity of using cerebral morphology for these purposes). Furthermore, the interpretation of some of the results seem to sidestep some crucial hypotheses in brain evolution.

The patterns found in humans are perhaps most relevant for the contribution the authors set out in the introduction. The results that are presented support previous work for expansion in cortical association areas along the human lineage. This is a pertinent result that suggests to me that the main emphasis of this work lies not with showing new insights on the evolution of human brain evolution, but in showing that cerebral morphology provides similar results compared with comparative neuroanatomical work. Perhaps this emphasis undersells the authors' contributions and more of an emphasis could be made on the new insights this approach provides.

One way to emphasize how this approach and these results could contribute to outstanding questions in human brain evolution is to compare the human trajectory to the trajectories in the great apes (and most crucially the trajectory in the ancestral lineage of the most recent common ancestor of great apes). Is the increase of motor, default mode and frontoparietal areas in the human lineage in line with such increase in other great apes? Or is it higher? This is an important debate that is not addressed here. This could be addressed by comparing such relative expansion patterns relative to the limbic system between the human lineage and lineages in the great apes (including the lineage leading to the most recent common ancestor of great apes). Is the rate of increase in motor/default/frontoparietal relative to limbic areas observed in humans on a par with those observed in other great ape lineages?

Adding such insights would increase an emphasis on how the approach and the type of data the authors present could contribute to ongoing questions in human brain evolution.

Another, more technical, issue relates to the fact that primates have evolved a unique suite of cortical association areas that are not present in non-primates. How did the authors deal with this when placing all species in a common reference space? I do appreciate that the pairwise surface matching that is employed provides some degree of buffer for the biases that this may create. But I see two intertwining issues that such pairwise matching would not buffer against: (1) the association between morphological landmarks and the anatomical position of cytoarchitectural areas is high in primary areas but low in association areas. I don't find any indication that this different degree of confidence is taken into account; (2) rodents simply don't have the same association areas as primates. Combining (1) and (2), how is the validity of comparing the rodents and the primates in a common reference space ensured?

I'm struggling to understand how this would not bias the results in some way. I appreciate that the authors seem to allude to this (though only indirectly) when highlighting the limitations of the study. But the potential impact of this should be described in more detail. If this is not expected to bias the results the authors should explain how; if this is expected to bias the results the authors should explain how this may affect the interpretation of the results.

A last comment refers to the use of the term 'mosaic'. In the literature this term has been very poorly defined and inconsistently used in the context of brain evolution. When it was introduced, it was suggested to refer both to a trend in which one brain area expands independently of other areas and a trend in which multiple brain areas expand together independently of other areas. In his influential

book "Principles of Brain Evolution", Georg Striedter pointed out the problems with such confusion of terms. He suggested that mosaic should only refer to the former trend, not to the latter. The latter is clearly better captured by the term 'modularity'; which is a term that has a long history in biology of meaning precisely this (i.e. a group/module of traits varying together independently of other traits). The use of the term 'mosaic' comes up several times in the manuscript and plays a crucial role in the interpretation of the results:

- "A mosaic of largely independent trajectories" (line 56)
- "A mosaic of significantly different expansion patterns" (line 176)
- "A mosaic-like modal segregation of primary cortical areas" (line 246)

In the first two instances, the term 'mosaic' seems to be used as a synonym for 'different' trajectories or expansion patterns. In both of these two instances the use of this term does not clearly align with either of the above-described meanings of this term. In the last instance, the authors seem to refer to the modular interpretation of the term (in which case it should be referred to as modular, not as mosaic). A more consistent use of this term is needed because the current use of this term confuses. If the authors do prefer to use the term 'mosaic' they should make it clear what they mean by it.

Reviewer #2:

Remarks to the Author:

This is an extremely ambitious manuscript that is the fruit of many years of work by a team of collaborative researchers. Its scope and ambitions are vast and so are the potential implications of the results. My main problem is that the manuscript is so dense that it becomes very hard to read and to follow. I must admit I believe the format is not appropriate for the range of information displayed. It is impossible to judge the paper without going deep into the supplementary methods. Although of course it is fine that details of the methods are not in the main text, at the moment I feel too much even of the conceptual framework behind the analyses is simply not explained. For instance, the main text suddenly talks about a 'common space', but nowhere in the main text is explained how this is created and how it is exploited. The main goal of the paper is also not quite clear after reading the introduction and the many implications of the study and its relationship to other recent comparative work hardly get a mention. Really, this is two or three papers condensed into a much too limited format.

I think the study has great potential, arguably better than Nature Communications, but it needs streamlining and made suitable for the highly multidisciplinary audience of the journal.

Because a number of my issues are due to an incomplete understanding of the methods, I highlight this. I need to see this addressed before I can judge the rest of the manuscript.

I must admit the techniques for the surface analyses are a bit beyond me. There are extensive references to field of surface processing and statistics, but I do believe a much more detailed description of the methods is required in the paper itself to make it sufficiently self-contained for a reader. As I have a reasonable background in the field, I expect they will be even more mysterious for quite a large part of the target audience of this paper. I would be very helpful if the conceptual framework can be made clearer and ideally if some figures can be produced that illustrate the pipeline in the sections from "Cortical shape encoding" onwards.

If I understand the method correctly, sister species are compared and the common ancestor is assumed to be half-way between those two brains. Is that correct? If so, this would ignore the parsimony in the comparative approach. For instance, it is widely assumed that the common ancestor of human and chimpanzee is more like a chimpanzee than a human, based on taking into account the next outgroup, the gorillas. But according to this approach it would be a half-human/half-chimpanzee? How is this addressed?

What is fundamentally unclear to me is how equivalence of regions across species is established. When claims are made about specific functional areas, I can see it is helpful to project this to the human brain. But surely to test that expansion is in these areas it is necessary to know where the equivalent areas (if any) are located in the rodent or the lagomorph? It is unclear to me how this problem is solved. As the authors state themselves "[earlier works] reveal a non-trivial relationship between anatomy and function".

One comment the authors make is that they use some additional measure to help the analyses, such as gyrification indices and manual labelling of the medial wall. What I cannot discern is whether that means gyrification is a scalar value and thus whether gyrification in a lagomorph is seen as equivalent to gyrification in a primate. The gyri of these two species are definitely not homologous, but I cannot figure out if that is an assumption of the model or not. This is likely to be even more problematic for the medial wall. Areas that are homologous are located in a very different location relative to the medial wall in rodents versus primates. Again, I am not sure how this would affect the conclusions.

The comparison of rodent and human cortical evolution is very nice, as it presents a way to judge the specificity of the method. However, again the sparse information means that many of the results remain unclear. First, could the authors show human and rodent expansions next to each other so that it's clear where the different species 'put their emphasis' in cortical evolution? The parcellation on Extended Fig. 4 is only interpretable with a human map, maybe that can be included in the figure. Again, here my not understanding how the regional equivalences between species is established makes it hard to judge the data.

I hope the authors can enlighten me, as I really would like to fully appreciate the manuscript.

Additional comments:

I don't agree that it's particularly the fact that we know have good computational renditions of cognition that makes the comparative approach particularly important, but it's up to the authors.

Unfortunately, the github link in the paper did not seem to work, so I could not see those results.

As a minor point, it might be nice to relate the common space as used here to other common space approaches used in the literature such as those used by Ting Xu and Nicole Eichert.

I'm not sure how habitat complexity can be ordered as arboreal to terrestrial to fossorial? Surely this is too much of a simplification? At the very least, I assume that arboreal is assumed to be the most complex? More justification of such decisions based on the ecological literature would be appreciated.

The species outlines on Figure 2 are a bit misleading. They are presumably meant to represent "last common ancestor of this species with human", but if the figure is reproduced (which one might expect it will) it will now suggest that the authors subscribe to a linear evolutionary great chain of being, which they don't. It also suggests the brain of *Ardipithecus* is in there, but there is no data in the study on subsections of the time since human and chimpanzee separation (~6 mya) and present humans.

Reviewer #3:

Remarks to the Author:

In this paper the authors collect and quantify a dataset on brain cortical surface morphology across Euarchontoglires. Through ancestral reconstruction and association analyses, they present evidence of an evolutionary link between cortex morphology and the "behavioral" ecological niche of species, focusing on the human lineage. The collected dataset is impressive and phylogenetically broad, and

the authors performed a huge amount of very thorough analyses. Although the question is not new and it has been tackled in many studies previously, the manuscript is of high potential interest for comparative evolutionary brain studies. Regrettably, I found the paper is not optimally written and it is hard to get through it with a clear understanding of what has been done and for what purpose. I think this is a disservice to the effort put on the analyses and data collection, so I think a considerable rewriting/editing would be required before publication.

Specifically, I found the introduction or the setting of the framework for the paper (first paragraph of the Main section) very short and omitting very important aspects, as giving a clear overview of the current state of knowledge, but most importantly, providing the reader with a clearly articulated rationale for the study. Hypotheses and the research questions are not clearly stated and what is indeed mentioned does not clearly follow logically from the broader context presented previously. In this sense, the results, being intertwined with description of methods, are hard to follow, with plenty of references to supplementary figures that makes reading and extracting the relevant information kind of hard. I suggest to more clearly state the question being answered with each analyses and to provide a more thorough or detailed description of the results. It seems to me that this manuscript was intended for another journal with very short space and submitted to this journal without appropriate reformatting. Another aspect is what I think is an excessive use of technical jargon that would make the paper hardly readable for someone that is not an expert, for example, on brain imaging. As this is a more general journal, I suggest to keep the technical jargon in the main body of the MS to a minimum, focusing instead on interpretability.

Regarding results,

- Suggestion: put the results of the analyses testing the ecological association before the ancestral reconstruction results (Fig 2), which I think are kind of complementary or descriptive and do not attempt to answer any clear specific hypotheses. A reorganization of results and rewriting to highlight the background and hypotheses being tested could make the paper more evolutionary and less descriptive, which would render it much more interesting and relevant.

- pPCAs related to ecological factors explain a very low percentage of the cortical variance. Could you please expand on this? How the conclusion is granted when explained variance is so low?

Regarding methods, I'm not familiar with all of them, so at this point I only have a few comments, but could potentially provide more extensive comments in a follow up review.

Minor comments:

At this point in the review process I'm not providing an extensive list of small suggestions that could be potentially changed if the authors considerably edit the manuscript, but here are a few that could help because the themes repeat all over.

Title

Perhaps the title is too broad. Cortical morphology in what organisms? What functional correlates? A more specific title could help the reader quickly get an idea of what the paper will be about exactly.

Abstract

"shape of the cerebral cortex"

please be more specific. Outer shape? This should be clearly defined.

Line 70-73 I think this paragraph should be moved to the previous section. Also, please state clearly

your research aims, some hypotheses, predictions. What and why are you doing this study? I think the introduction also does a poor job describing what is known and what is not regarding the problem the authors are tackling. This section needs expansion to clearly reflect the current knowledge and a clearly stated research question.

Line 74 What does the convex hull represent, or what is it? This is first mentioned here so please be more precise.

Line 79 "(...) we estimated the evolution of the area of the cortical surfaces (...)"

What does it mean to "estimate the evolution", please be more precise.

Line 85 "We iterated these steps, progressing upwards in the tree and upon reaching the root node resampled all surfaces to a common topology"

I don't understand what to "resample all surfaces to a common topology" means. Could you clarify?

Figure 1 legend "The resulting surface models are compared 64 with endocasts (Supplementary Table 1b) obtained from phylogenetically closely related 65 fossil endocasts at the branching points indicated by the letters A-I"

There are no branching points (nodes) indicated on the phylogeny. Please correct.

Figure 2 – Labels are missing everywhere in the plots, it's very hard to read. Please use letters to subdivide the figure and make it easier to reference each part in the legend.

For example "The y-axis in the plot showing the area-increase of clusters, also denotes the relative area across the cluster on the surface" could be simplified with e.g "b) area-increase of clusters. Y-axis denotes the relative area across..." or similar.

Line 120 " The maxima of association between expansion and functional categories of the components revealed a sequence of neuromorphological stages, starting with the expansion of visual areas, followed by parahippocampal regions in the late Cretaceous, auditory and sensory-motor areas in the early Oligocene until the mid-Miocene and finally expansion of association areas in the late Miocene and Pliocene up until the present day."

Here I got lost. The section starts describing morphological changes in the "human lineage"/Homo but in the following paragraph you describe changes happening in the late Cretaceous? Primates as a whole barely existed by then. Please clarify.

- Each section describes analyses but without clearly stating the purpose of such analyses. It is very hard to follow the train of thought of the authors. Please provide clear context and concise aims.

Line 128 "We then compared the evolutionary expansion patterns in the two distinct lineages of mice and humans (Extended Data Figure 4) as they represent ubiquitously studied species of rodents and primates."

Why? Just because they are very studied is not enough to justify studying them more. Again, please be more precise with the research question you are trying to answer. Guide the reader through the analyses to understand what was done and for what reason.

Line 159 "We performed phylogenetic principal component analysis (pPCA) to test if ecological and

behavioural niches are reflected in the shape of the cerebral cortex.”

PCA of what data? PCA is a dimensionality reduction technique, it's not for testing association with other variables. You reduced the number of variables using pPCA and then tested for the association of this variables with the niche using another method. Please clarify.

Line 170 “By mapping the relative expansion patterns into a functionally defined parcellation of the human cerebral cortex, we found significant group differences (...)”

What groups? I couldn't follow, could you please clarify?

Extended Data Figure 2 – In the plots, the correct name is “Primates”, not “Primata”
What is the scale at the right of the plot showing?

Methods

Line 343 “Species that were reported as either diurnal, cathemeral or arrhythmic in the literature were labelled as “diurnal”.

I'm not familiar with all studied species to provide an assessment of the validity of this simplification, but for example it is known that the platyrrhine *Aotus* (Night/Owl monkey) despite being “cathemeral” has extensive adaptations to this niche, for example, enlarged orbits that could potentially be reflected in cortical morphology. I'm not asking the author's to repeat all the analyses (probably results won't change as cathemerality in *Aotus* is a derived condition) but a line justifying why the simplification is granted would be useful.

Line 508 – “However, the scalar formulation of mixture of OU models is defined only for ultrametric trees (eg. trees whose tips all have the same age). We therefore shift the tip ages for the fossil data to the present day. For the extinct hominin species included in the augmented tree, only estimates of the SA of the convex hull of the brain are available from endocasts of the fossil crania; the SA of the cerebral cortex is considered missing in phylogenetic modelling. We assume that the slight misspecification of the tip-dates for the fossil hominin data is immaterial to the relative BIC values used for model selection. ”

I would argue the whole point of incorporating fossil species is that they predate extant species, so forcing them to be at the present make their use at the bare minimum pointless, and more likely, distortive of the results. Did you do this only for hominin fossils? If this is the case I could be ok with that although I see no point. But if older species were forced to be at present, this could clearly distort OU models because variance could be potentially inflated, thus impacting the model parameters (sigma, alpha and/or the optimum).

We want to sincerely thank the reviewers for taking the time to provide thoughtful and supportive comments on our paper. We feel that we were able to considerably improve the paper by addressing the issues that have been raised. In the following, we provide point by point answers to each of them.

REVIEWER COMMENTS

Reviewer #1 (Remarks to the Author):

R1-1:

I should note at the start of this review that I'm not an expert in morphological/spatial statistics (so I cannot evaluate the validity of these aspects of the paper), though I am an expert in phylogenetic comparative methods and brain evolution.

In my view, this article presents the most convincing case to date that "ecological and behavioral adaptations during evolution are mirrored in the shape of the cerebral cortex across extant species". However, the balance of emphasis of the paper perhaps weighs too heavily on making this point, and too lightly on emphasizing the new insights these results may provide.

A: We thank the reviewer for the encouraging comment. Indeed, the paper focuses on contributing evidence for the relationship between the evolution of the cerebral cortex, and ecological and behavioural adaptations. Aside from this observation, the quantitative and statistical results presented in this paper should also serve to validate the underlying "common reference frame", i.e., the dataset of aligned surface models for all 90 species studied in the paper, which will accompany its publication. We hope that this resource will serve other researchers with specific questions in palaeontology, paleoneurology, comparative neurosciences or translational medicine to test and answer their specific research questions within an evolutionary framework. In addition to the findings reported in the paper, we now clarify how these results may be used to further such insights. We have changed the title of the manuscript to "*Evolution of Cortical Geometry and its Link to Function, Behaviour and Ecology*", rewritten the introduction and added a short conclusion section after the discussion to clarify the intent of the paper:

"Here, we provide a unique, publicly available dataset and atlas that allows us to trace fundamental aspects of the deep evolutionary history of the shape of the human cerebral cortex. We used it to characterise morphological changes from the expansion of occipital areas due to increasing reliance on visual information in Eutherian ancestors of early Euarchontoglires some 77Mya until the globularisation of the cortex due to expanding association areas in hominids 4Mya. We show that these evolutionary processes can be decomposed into individual components with concise functional interpretations and relationships to ecology. Importantly, we demonstrate the spatial heterogeneity and inter-species variability of these processes, highlighting the importance of considering the evolutionary origins of the structures and functions studied in neuroscience, and the resulting inherent intricacies in translating experimental results between species. Future research may benefit from the availability of a common neuroanatomical reference space for this important part of the mammalian class." (Line 594ff)

R1-2:

Overall, the results are intuitive and confirm what one would expect both for human brain evolution and primate brain evolution compared with rodent brain evolution. The clarity of the results are impressive, but I'm not wow-ed by landmark new findings (aside from demonstrating

the validity of using cerebral morphology for these purposes). Furthermore, the interpretation of some of the results seem to sidestep some crucial hypotheses in brain evolution.

A: Thank you, we have clarified the purpose of the paper in the introduction:

“Here, we link phylogenetic and geometric methods to establish a common reference frame for the cerebral cortex of Euarchontoglires, the supra-order containing both humans as well as commonly used neuroscientific model animals such as mice, rats, macaques and marmosets. By establishing group-wise correspondences between all cortical surfaces in a dataset of 90 different species, we show how ecological adaptations are reflected in the shape of the surfaces of cerebral cortices of living members of Euarchontoglires. We use spatial statistics to quantify the process of modal segregation of brain function. Finally, we combine ancestral state reconstruction with statistical decoding of brain activation to derive a historical timeline of modules of evolutionary adaptations of the cortical shape in the deep ancestral human lineage since the last common ancestor of rodents and primates. The resulting aligned surface meshes for all extant species, their coordinates for commonly used template spaces as well as their estimated ancestral shapes are available at <https://github.com/MrBurnst/EvolutionOfCorticalShape>. “ (Line 81ff)

The aim of the paper is twofold. First, it demonstrates that a holistic analysis of the phylogeny of cerebral morphology can contribute evidence regarding previous hypotheses on brain evolution such as the tethering hypothesis, the social brain hypothesis, or the dominant role of visual cognition in primate brain evolution. The second main contribution is a common reference and mapping technique across a multitude of species, which can enable comparative neuroscience studies to investigate the evolution of concrete aspects of neuroanatomy and function, as well as to evaluate the usefulness/appropriateness of certain animal models for specific research questions in humans. Importantly, the data and resources of this evolutionary framework are shared openly with the scientific community, providing a tool to advance comparative research.

R1-3:

The patterns found in humans are perhaps most relevant for the contribution the authors set out in the introduction. The results that are presented support previous work for expansion in cortical association areas along the human lineage. This is a pertinent result that suggests to me that the main emphasis of this work lies not with showing new insights on the evolution of human brain evolution, but in showing that cerebral morphology provides similar results compared with comparative neuroanatomical work. Perhaps this emphasis undersells the authors’ contributions and more of an emphasis could be made on the new insights this approach provides.

One way to emphasize how this approach and these results could contribute to outstanding questions in human brain evolution is to compare the human trajectory to the trajectories in the great apes (and most crucially the trajectory in the ancestral lineage of the most recent common ancestor of great apes). Is the increase of motor, default mode and frontoparietal areas in the human lineage in line with such increase in other great apes? Or is it higher? This is an important debate that is not addressed here. This could be addressed by comparing such relative expansion patterns relative to the limbic system between the human lineage and lineages in the great apes (including the lineage leading to the most recent common ancestor of great apes). Is the rate of increase in motor/default/frontoparietal relative to limbic areas observed in humans on a par with those observed in other great ape lineages?

Adding such insights would increase an emphasis on how the approach and the type of data the authors present could contribute to ongoing questions in human brain evolution.

A: We agree that such a study would be highly informative and made possible by the proposed technique. In the interest of space, and to allow for a full description of the general methodology, we only included one additional comparative analysis in Extended Data Figure 4, the comparison of the expansion of functionally defined regions in the human and mice lineage:

Extended Data Figure 4: (a) Estimated oxygen concentration in comparison with correlation between cortical surface area of mice and humans indicates influence of climatic changes to diverging evolutionary adaptation ($r=0.915$, $p<0.001$, 0.95 CI [0.873, 0.944], Supplementary Table 8a). (b) Functional parcellation of the cerebral cortex obtained from human fMRI data plotted on the ancestral state reconstruction of the cortical surface model of the LCA of rodents and primates (c) Phylogenetic tree of the species of Euarchontoglires annotated with the relative extent of functional areas. (d) Comparison of the changes in relative area attributed to individual functionally defined regions of the cortex shows a sequence of potential evolutionary processes and highlights the distinctive reduction of the surface area attributed to limbic processes and the relative

expansion of association areas in *Homo*. In *Mus* on the contrary, these regions dramatically expanded, mostly but not exclusively attributable to a dramatically expanded olfactory bulb. Corresponding statistical information is provided in Supplementary Table 8b. **(e)** Comparison between estimated relative local evolutionary cortical surface expansion in mice (top) and humans (bottom)

Having said that, the proposed comparison is of course highly relevant, and we would like to investigate this in separate work.

R1-4:

Another, more technical, issue relates to the fact that primates have evolved a unique suite of cortical association areas that are not present in non-primates. How did the authors deal with this when placing all species in a common reference space? I do appreciate that the pairwise surface matching that is employed provides some degree of buffer for the biases that this may create. But I see two intertwining issues that such pairwise matching would not buffer against: (1) the association between morphological landmarks and the anatomical position of cytoarchitectural areas is high in primary areas but low in association areas. I don't find any indication that this different degree of confidence is taken into account; (2) rodents simply don't have the same association areas as primates. Combining (1) and (2), how is the validity of comparing the rodents and the primates in a common reference space ensured?

I'm struggling to understand how this would not bias the results in some way. I appreciate that the authors seem to allude to this (though only indirectly) when highlighting the limitations of the study. But the potential impact of this should be described in more detail. If this is not expected to bias the results the authors should explain how; if this is expected to bias the results the authors should explain how this may affect the interpretation of the results.

A: We appreciate this issue. It points to an important aspect of mapping across the cortical geometry of different species that needed to be explained in more detail in the paper. The diffeomorphic mapping between the cortical surfaces of different species explains emerging areas by local diffeomorphic expansion of a cortical region close to the new area, driven by geometric similarity of the surrounding vicinity and the overall localization.

However, as pointed out by the reviewer, the choice of a diffeomorphic correspondence model entails the limitation of the absence of cytological or functional de-novo cortical areas across species. The present paper does not aim to answer the question why different aspects of specific anatomical areas do not correspond, notably in evolutionary 'younger' areas such as association areas. Different alignments have been proposed such as based on function (Xu 2020) or structural connectivity (Eichert 2020, Mars 2018). The results of these different approaches to alignment do not necessarily agree, and the question on where and why this is the case is likely the most interesting question in (evolutionary) neuroscience (Mars 2021).

As has been proposed (Mars 2021), the comparison of different alignments based on specific modalities should prove a very fruitful avenue of future research. In this work, we demonstrated

that global morphology can serve as a feature for establishing one possible alignment, and that this approach leads to results that are in line with established knowledge on neuroevolution from other sources. However, we do not aim to suggest that it is the only 'correct' common reference in which the cortices of different species should be compared, as alignments based on different modalities will surely disagree in different areas to varying degrees. Future work analysing these differences appears likely to inform neuroevolutionary questions.

Furthermore, it should be noted that we made the conscious choice to aim for a dense sampling of Euarchontoglires due to the relevance of species of the clade in neuroscience research. By obtaining a dense sampling, the results of pairwise matching can - on average - be considered more 'valid' as the time for species-specific divergences is shorter. Unfortunately, there does not yet exist a comparable dataset of Euarchontoglires in any other structural imaging modality (we do not have access to the Mami database of (Assaf 2020). but combination of ours and their results would be a fantastic future research project). Leveraging functional data to inform cross-species comparisons is limited by both availability and challenges in task selection and acquisition. In functional data on the other hand, there also does on the one hand not exist any comparable dataset, and there is the additional question of establishing correspondence between functional regions when the functional equivalence of the current state of the cortex (be it at rest, either sedated or not, or while performing a task) is even more difficult to establish across species.

Xu, T., Nenning, K. H., Schwartz, E., Hong, S. J., Vogelstein, J. T., Goulas, A., ... & Langs, G. (2020). Cross-species functional alignment reveals evolutionary hierarchy within the connectome. *Neuroimage*, 223, 117346.

Eichert, Nicole, Emma C. Robinson, Katherine L. Bryant, Saad Jbabdi, Mark Jenkinson, Longchuan Li, Kristine Krug, Kate E. Watkins, and Rogier B. Mars. "Cross-species cortical alignment identifies different types of anatomical reorganization in the primate temporal lobe." *Elife* 9 (2020): e53232.

Mars, R. B. et al. Whole brain comparative anatomy using connectivity blueprints. *Elife* 7, (2018).

Mars, R. B., Jbabdi, S. & Rushworth, M. F. S. A Common Space Approach to Comparative Neuroscience. *Annu. Rev. Neurosci.* 44, 69–86 (2021).

Assaf, Y., Bouznach, A., Zomet, O., Marom, A. & Yovel, Y. Conservation of brain connectivity and wiring across the mammalian class. *Nature Neuroscience* vol. 23 805–808 (2020)

R1-5:

A last comment refers to the use of the term 'mosaic'. In the literature this term has been very poorly defined and inconsistently used in the context of brain evolution. When it was introduced, it was suggested to refer both to a trend in which one brain area expands independently of other areas and a trend in which multiple brain areas expand together independently of other areas. In his influential book "Principles of Brain Evolution", Georg Striedter pointed out the problems with

such confusion of terms. He suggested that mosaic should only refer to the former trend, not to the latter. The latter is clearly better captured by the term 'modularity'; which is a term that has a long history in biology of meaning precisely this (i.e. a group/module of traits varying together independently of other traits).

The use of the term 'mosaic' comes up several times in the manuscript and plays a crucial role in the interpretation of the results:

- "A mosaic of largely independent trajectories" (line 56)
- "A mosaic of significantly different expansion patterns" (line 176)
- "A mosaic-like modal segregation of primary cortical areas" (line 246)

In the first two instances, the term 'mosaic' seems to be used as a synonym for 'different' trajectories or expansion patterns. In both of these two instance the use of this term does not clearly align with either of the above-described meanings of this term. In the last instance, the authors seem to refer to the modular interpretation of the term (in which case it should be referred to as modular, not as mosaic). A more consistent use of this term is needed because the current use of this term confuses. If the authors do prefer to use the term 'mosaic' they should make it clear what they mean by it.

A: We agree that the term was not used consistently in the manuscript and have reduced and clarified its use. We use the term as intended by G. Striedter to highlight the dichotomy of heterogeneous evolution of individual aspects and entirely concerted evolution in the introduction of the paper but removed any other reference in the remainder of the text. Instead, we discuss our results as "independent modules" or "different expansion patterns".

The three sentences now read:

- *"The heterogeneous change of the extent and amount of individual cortical areas during evolution has spurred a debate on whether these changes of function and topography occurred in a concerted manner or are better described as a **mosaic of dynamically changing areas**" (Line 59ff)*
- *"Post-hoc tests revealed a mosaic of significantly differing expansion patterns associated with both ecological and behavioural variables" changed to "Post-hoc tests revealed significantly differing expansion patterns associated with both ecological and behavioural variables." (Line 208)*
- *"Together, these results indicate that in accordance with the tethering hypothesis, a mosaic-like modal segregation of primary cortical areas increased during evolution in the human lineage" changed to "In accordance with the tethering hypothesis, our results show that the modular segregation of primary cortical areas increased during human evolution" (Line 506ff)*

Reviewer #2 (Remarks to the Author):

R2-1:

This is an extremely ambitious manuscript that is the fruit of many years of work by a team of collaborative researchers. Its scope and ambitious are vast and so are the potential implications of the results. My main problem is that the manuscript is so dense that it become very hard to read and to follow. I must admit I believe the format is not appropriate for the range of information displayed. It is impossible to judge the paper without going deep into the supplementary methods. Although of course it is fine that details of the methods are not in the

main text, at the moment I feel too much even of the conceptual framework behind the analyses is simply not explained. For instance, the main text suddenly talks about a ‘common space’, but nowhere in the main text is explained how this is created and how it is exploited. The main goal of the paper is also not quite clear after reading the introduction and the many implications of the study and its relationship to other recent comparative work hardly get a mention. Really, this is two or three papers condensed into a much too limited format. I think the study has great potential, arguably better than Nature Communications, but it needs streamlining and made suitable for the highly multidisciplinary audience of the journal.

A: Thank you for this comment. We have clarified the two main aims of the paper in the introduction and have worked through-out the entire manuscript to provide more structure, and to ensure that any methods are explained before they are referred to. For this we have also introduced brief explanations into the main text, where more detailed descriptions are available in the supplementary material.

The two main aims of the paper are first to provide evidence for the link between the evolution of cortical morphology and evolutionary adaptations to ecological niches, second to provide a common reference frame as a tool for other researchers, that enables the mapping of cortical features across a large number of extant species in comparative neuroscience.

The introduction now reads as follows:

“Here, we link phylogenetic and geometric methods to establish a common reference frame for the cerebral cortex of Euarchontoglires, the supra-order containing both humans as well as commonly used neuroscientific model animals such as mice, rats, macaques and marmosets. By establishing group-wise correspondences between all cortical surfaces in a dataset of 90 different species and show how ecological adaptations are reflected in the shape of the surfaces of cerebral cortices of living members of Euarchontoglires. We use spatial statistics to quantify the process of modal segregation of brain function. Finally, we combine ancestral state reconstruction with statistical decoding of brain activation to derive a historical timeline of modules of evolutionary adaptations of the cortical shape in the deep ancestral human lineage since the last common ancestor of rodents and primates. The resulting aligned surface meshes for all extant species, their coordinates for commonly used template spaces as well as their estimated ancestral shapes are available at <https://github.com/MrBurnst/EvolutionOfCorticalShape>. “ (Line 81ff)

The corresponding part of the discussion now reads as follows:

“In this study, we established a common reference frame for the study of the cerebral cortex in Euarchontoglires from 90 different species. We used the resulting dense correspondences between cortical surfaces to reconstruct the evolution of the cortical shape along the deep ancestral human lineage. We demonstrated how ecological and behavioural adaptations during evolution were mirrored in the shape of the cerebral cortex across extant species. Neuroanatomical correlates of the resulting socio-ecological niches of animals are still present in the variations of the shape of the cerebral cortex in extant species. Furthermore, the complexity of both the ecological and social environment is associated with the selective expansion of functionally specific cortical regions. Using spatial statistics, we demonstrated a gradual evolutionary decrease in the spatial range at which the modal specificity of each of these cortical areas becomes independent of the others suggesting the emergence of areas with multiple co-located associations. This corroborates the tethering hypothesis, the evolutionary process of concentration of mono-modal areas alongside an expansion

of multimodal areas between them. By decoding the neurological function of the estimated cortical expansion in the deep ancestral human lineage using meta-analytic functional maps, we extracted both temporally and spatially circumscribed neuromorphological events that shaped the cerebral cortex of eutherians including Euarchontoglires over the past 77 million of years.” (Line 358ff)

R2-2:

Because a number of my issues are due to an incomplete understanding of the methods, I highlight this. I need to see this addressed before I can judge the rest of the manuscript.

I must admit the techniques for the surface analyses are a bit beyond me. There are extensive references to field of surface processing and statistics, but I do believe a much more detailed description of the methods is required in the paper itself to make it sufficiently self-contained for a reader. As I have a reasonable background in the field, I expect they will be even more mysterious for quite a large part of the target audience of this paper. I would be very helpful if the conceptual framework can be made clearer and ideally if some figures can be produced that illustrate the pipeline in the sections from “Cortical shape encoding” onwards.

A: We agree with the reviewer that an overview of the methods used to arrive at the results presented in the paper is necessary for the readers. We have worked on providing compact but informative explanations in the text and have added a figure (Figure 2) that summarises the crucial technical aspects of our methodological workflow, together with a summary of the main steps at the beginning of the method section.

Figure 2: Methodological overview. Overview of the steps required to perform shape alignment and ancestral state reconstruction between the surface models of the cerebral cortices of two sister species. (a) For two species, anatomy and geometric features are assigned to each point on their cortical mesh; (b) after spectral embedding of both surfaces, alignment establishes dense correspondence between the cortices of the two species; (c) the shape of the common ancestors is reconstructed based on these correspondences

by interpolating in the space of smooth shells; (d) finally, cortical features are estimated for the common ancestor, and the reconstruction is iterated until the root of the phylogenetic tree is reached.

At the same time, we have kept the more detailed explanation of the individual steps in the supplemental material, and hope that this is a helpful structure for the readers.

R2-3:

If I understand the method correctly, sister species are compared and the common ancestor is assumed to be half-way between those two brains. Is that correct? If so, this would ignore the parsimony in the comparative approach. For instance, it is widely assumed that the common ancestor of human and chimpanzee is more like a chimpanzee than a human, based on taking into account the next outgroup, the gorillas. But according to this approach it would be a half-human/half-chimpanzee? How is this addressed?

A: We apologise that this has not been conveyed more clearly in the manuscript. Due to space limitations, much of the methods had to be relegated to the supplementary material. We did not assume a half-way approximation. Estimates of divergence times were used as part of the method. We added a figure (Figure 2c) in which we summarise the methods used. Furthermore, we clarify the approach by which ancestral state reconstruction is performed in the first section of the results:

"We then performed pairwise matching between the cortical surfaces of sister species and approximated their ancestral state by interpolating in the space of smooth shells, where we used the relative differences in the global shape features to determine the interpolation factor. The relative position of the estimated ancestral and observed daughter species in the nonlinear space of smooth shells thus corresponds to the euclidean distance between the corresponding global shape features."
(Line 134ff)

More details on the phylogenetic model are given in the methods under "Phylogenetic modelling" (Line 820ff) and in particular for the point raised by the reviewer in the sub-section "Phylogenetic modelling of global cerebral morphology" (Line 833ff). Finally, a mathematical presentation of the relevant methods is given in the Supplementary Methods under Section 1, "Phylogenetic Modelling" as well as Section 3.2.2, "Estimation of interpolation factors between sister species".

R2-4:

What is fundamentally unclear to me is now equivalence of regions across species is established. When claims are made about specific functional areas, I can see it is helpful to project this to the human brain. But surely to test that expansion is in these areas it is necessary to know where the equivalent areas (if any) are located in the rodent or the lagomorph? It is unclear to me how this problem is solved. As they authors state themselves "[earlier works] reveal a non-trivial relationship between anatomy and function".

One comment the authors make is that they use some additional measure to help the analyses, such as gyrification indices and manual labelling of the medial wall. What I cannot discern is whether that means gyrification is a scalar value and thus whether gyrification in a lagomorph is seen as equivalent to gyrification in a primate. The gyri of these two species are definitely not homologous, but I cannot figure out if that is an assumption of the model or not. This is likely to be even more problematic for the medial wall. Areas that are homologous are location in a very different location relative to the medial wall in rodents versus primates. Again, I not sure how this would affect the conclusions.

A: We regret that this has not been conveyed more clearly in the paper. We use the overall geometry and gross anatomical features such as the extent of the medial wall and corpus callosum (which are strongly established homologies in the clade of Euarchontoglires). The surface matching exploits geometrical features at different scales (sulci, gyri) where they are available for the alignment of two sister species. We have added a figure (Figure 2) to summarise the methodological workflow. We hope that this facilitates understanding and appreciation of the results, as we would like to refrain from adding more methodological details to the body in the interest of readability.

Figure 2: Methodological overview. Overview of the steps required to perform shape alignment and ancestral state reconstruction between the surface models of the cerebral cortices of two sister species. (a) For two species, anatomy and geometric features are assigned to each point on their cortical mesh; (b) after spectral embedding of both surfaces, alignment establishes dense correspondence between the cortices of the two species; (c) the shape of the common ancestors is reconstructed based on these correspondences by interpolating in the space of smooth shells; (d) finally, cortical features are estimated for the common ancestor, and the reconstruction is iterated until the root of the phylogenetic tree is reached.

Details on how correspondences between shapes are established can be found in the methods section (under “cortical shape encoding”, Line 696 and following), while a mathematically rigorous explanation is provided as Supplementary Methods under Section 3 - “Cortical surface matching”.

Briefly, the method used in the paper uses differential geometry to encode features describing the shape of the cortex at different scales. We perform an eigendecomposition of the cotangent

Laplacian matrix corresponding to the cortical surface models of two sister species (augmented by previously described additional anatomical information). Then these three-dimensional models are projected onto a high-dimensional space in which alignment can be performed using tools from computer vision and shape analysis.

Importantly, this process does not assume functional correspondences as a basis for registration. Instead, the paper demonstrates that the analysis of the correspondences obtained by using only geometric information of cortical surfaces recovers information on evolutionary adaptation to ecological niches as well as a timeline of human brain evolution that - importantly - closely matches established knowledge from other sources in comparative neuroanatomy.

R2-5:

The comparison of rodent and human cortical evolution is very nice, as it presents a way to judge the specificity of the method. However, again the sparse information means that many of the results remain unclear. First, could the authors show human and rodent expansions next to each other so that it's clear where the different species 'put their emphasis' in cortical evolution? The parcellation on Extended Fig. 4 is only interpretable with a human map, maybe that can be included in the figure. Again, here my not understanding how the regional equivalences between species is established makes it hard to judge the data. I hope the authors can enlighten me, as I really would like to fully appreciate the manuscript.

A: We agree with the reviewer that direct visualisation of the estimated expansions is helpful in reducing the anthropocentricity of the results visualised in Extended Data Figure 4. We had in fact already computed these visualisations but chose not to include them as the shape of the cortex of mice (and rodents in general) shows little evolutionary variability compared to primates and especially humans. Therefore, colour-coding the relative surface expansion is somewhat misleading, as it over-emphasises small changes that might be attributable to dataset bias or inherent numerical limitations in the discretization of the surfaces and their alignment. We have added an additional panel comparing the local relative evolutionary expansion of the cortical surface in the human lineage (which is already shown in Figure 7 in the main text) and that of the mice lineage. Despite the aforementioned difficulties, one can appreciate the expansion of the olfactory bulb in the lineage of mice, as can also be observed in panel d of the same figure in the "limbic" are colour-coded in light beige.

Extended Data Figure 4: (a) Estimated oxygen concentration in comparison with correlation between cortical surface area of mice and humans indicates influence of climatic changes to diverging evolutionary adaptation ($r=0.915$, $p<0.001$, 0.95 CI [0.873, 0.944], Supplementary Table 8a). (b) Functional parcellation of the cerebral cortex obtained from human fMRI data plotted on the ancestral state reconstruction of the cortical surface model of the LCA of rodents and primates (c) Phylogenetic tree of the species of Euarchontoglires annotated with the relative extent of functional areas. (d) Comparison of the changes in relative area attributed to individual functionally defined regions of the cortex shows a sequence of potential evolutionary processes and highlights the distinctive reduction of the surface area attributed to limbic processes and the relative expansion of association areas in *Homo*. In *Mus* on the contrary, these regions dramatically expanded, mostly but not exclusively attributable to a dramatically expanded olfactory bulb. Corresponding statistical information is provided in Supplementary Table 8b. (e) Comparison between estimated relative local evolutionary cortical surface expansion in mice (top) and humans (bottom)

Additional comments:

R2-6:

I don't agree that it's particularly the fact that we know have good computational renditions of cognition that makes the comparative approach particularly important, but it's up to the authors.

A: We agree with the reviewer that this comment is out of place in this article and have removed it.

R2-7:

Unfortunately, the github link in the paper did not seem to work, so I could not see those results.

A: Due to the double-blind review process, we could not find a way of sharing this data maintaining anonymity. The github link will become active upon publication.

R2-8:

As a minor point, it might be nice to relate the common space as used here to other common space approaches used in the literature such as those used by Ting Xu and Nicole Eichert.

A: We agree with the reviewer that such a comparison would be highly valuable and are considering performing a proper comparison of mappings based on different modalities in future work. However, we find the present paper already very dense as it stands and avoided adding more results so as to not dilute the main points of our results. While a detailed discussion of the differences between alignments obtained from different modalities is beyond the scope of this paper, we of course cite these and other relevant publications in comparative computational neuroanatomy. We consider such a comparison a highly valuable future direction of research that could be conducted using the data published with this manuscript.

R2-9:

I'm not sure how habitat complexity can be ordered as arboreal to terrestrial to fossorial? Surely this is too much of a simplification? At the very least, I assume that arboreal is assumed to be the most complex? More justification of such decisions based on the ecological literature would be appreciated.

A: We base the ordering of the habitat less on a-priori assumptions about their nature but rather on the overall correlation of the relative expansion profiles associated with them. Figure 4 in the revised version of the manuscript explains this now in more detail. However, the observation of an ordinal nature of habitat does not directly entail a relationship with complexity, and we agree with the reviewer that that notion is very contentious. While we follow Betrand 2021 (<https://doi.org/10.1038/s42003-021-01887-8>) in assuming a higher "complexity" for arboreal in contrast to terrestrial and fossorial environments, this represents a stark simplification of a multitude of ecological factors.

Nonetheless, we consider due to increased cognitive demands posed by arboreal environments for perception, proprioception, locomotion as well as affordance selection, the importance of attention as well as spatial memory, the proposed ordering to be valid, albeit simplified. Under the affordance competition hypothesis (Cisek 2007), brain evolution is related to the availability of an increased number of possible ways of action ("affordances"), both in terms of action (eg. action maps, complex limb movements) as well as perception (eg. due to an increase of perceptive distance, and/or sensory stimuli present in the environment (shapes, scents, sounds, combinations thereof). Arboreal habitats represent an increase in both of these dimensions. We interpret our results of evolutionary expansion of the dorsal compared to the ventral aspects of the cortex in that line in Line 480ff:

“An alternative or possibly complementary explanation for a relationship between the expansion of areas of the dorsolateral cortex and complexity of a species’ environment can be found in the affordance competition theory of cortical organisation. Under this framework, a selective expansion of areas of the dorsal stream in complex environments could lead to an increased complexity of proposed potential modes of action such as action maps. Since the complexity of selecting amongst them however stays comparatively constant, no expansion of the ventral stream is required.”

Thus, this stratification allows us to represent broad differences of cognitive demand. Complexity is one part, but other fitness factors such as injury risk that drive neuromuscular diversification (e.g. prehensile tails and gliding) and demand greater precision of neuromuscular integration make the arboreal categorisation an important reference point for comparisons against terrestrial and fossorial habitats. Similar levels of ecological stratification also used in for example:

Harvey, P. H., Clutton-Brock, T. H., & Mace, G. M. (1980). Brain size and ecology in small mammals and primates. *Proceedings of the National Academy of Sciences*, 77(7), 4387-4389.

Bernard, R. T. F., & Nurton, J. (1993). Ecological correlates of relative brain size in some South African rodents. *African Zoology*, 28(2), 95-98.

Cisek, Paul. "Cortical mechanisms of action selection: the affordance competition hypothesis." *Philosophical Transactions of the Royal Society B: Biological Sciences* 362, no. 1485 (2007): 1585-1599.

R2-10:

The species outlines on Figure 2 are a bit misleading. They are presumably meant to represent “last common ancestor of this species with human”, but if the figure is reproduced (which one might expect it will) it will now suggest that the authors subscribe to a linear evolutionary great chain of being, which they don’t. It also suggests the brain of Ardipithecus is in there, but there is no data in the study on subsections of the time since human and chimpanzee separation (~6 mya) and present humans.

A: The silhouettes in Figure 7 (Figure 2 in the original version of the manuscript) correspond to artistic renditions of the estimated bodily morphology of the known extinct species that, based on current paleontological knowledge, is evolutionary closest to the point in the deep ancestral human lineage for which we estimated the shape of the cortical surface based on our data (shown in the upper part of the same figure). We do not intend to suggest that any fossil data was included in the dataset. Rather, these drawings should help to illustrate the approximate overall morphology (eg. quadrupedal or bipedal), mode of locomotion and habitat of a possible ancestor at that time. We include these figures to give readers with little or no background in palaeontology an intuition of the estimated ecology of our ancestors at a given point in evolution and help put the results based on the estimated evolutionary expansion of the cortex presented in the figure in context.

Reviewer #3 (Remarks to the Author):

R3-1:

In this paper the authors collect and quantify a dataset on brain cortical surface morphology across Euarchontoglires. Through ancestral reconstruction and association analyses, they present

evidence of an evolutionary link between cortex morphology and the “behavioral” ecological niche of species, focusing on the human lineage. The collected dataset is impressive and phylogenetically broad, and the authors performed a huge amount of very thorough analyses. Although the question is not new and it has been tackled in many studies previously, the manuscript is of high potential interest for comparative evolutionary brain studies. Regrettably, I found the paper is not optimally written and it is hard to get through it with a clear understanding of what has been done and for what purpose. I think this is a disservice to the effort put on the analyses and data collection, so I think a considerable rewriting/editing would be required before publication. Specifically, I found the introduction or the setting of the framework for the paper (first paragraph of the Main section) very short and omitting very important aspects, as giving a clear overview of the current state of knowledge, but most importantly, providing the reader with a clearly articulated rationale for the study.

A: We agree with the reviewer that the original version of the manuscript was too dense and thus difficult to follow. We initially decided to split the contents of the manuscript into a compact main article to convey the findings in a compact form while relegating both methodological details as well as detailed interpretations to separate supplementary documents.

We understand that this format made the main text too difficult to follow and caused confusion regarding the aims of the paper. We have thus restructured the manuscript and provide a more self-contained description of the methods in the main paper.. The mathematical details of the methods remain in the supplemental material. We now also include the discussion of the results in the main text and restructured it to make it more concise and easier to follow.

A:

R3-2:

Hypotheses and the research questions are not clearly stated and what is indeed mentioned does not clearly follow logically from the broader context presented previously. In this sense, the results, being intertwined with description of methods, are hard to follow, with plenty of references to supplementary figures that makes reading and extracting the relevant information kind of hard. I suggest to more clearly state the question being answered with each analyses and to provide a more thorough or detailed description of the results. It seems to me that this manuscript was intended for another journal with very short space and submitted to this journal without appropriate reformatting.

A: We agree with the reviewer that the initial version of the manuscript was very condensed and at times difficult to follow. We restructured it substantially to make its intent and purposes clear. We rewrote the introduction as well as merged the supplementary into the main discussion, yielding a more concise interpretation of our results. This restructuring in turn allowed us to move Figure 4 (originally Extended Data Figure 6), further increasing interpretability. We rewrote large parts of the results section as well, trying to increase readability by supplying motivation for each subsection.

R3-3:

Another aspect is what I think is an excessive use of technical jargon that would make the paper hardly readable for someone that is not an expert, for example, on brain imaging. As this is a more general journal, I suggest to keep the technical jargon in the main body of the MS to a minimum, focusing instead on interpretability.

A: We agree with the reviewer that due to the complexity of the subject area and the methods employed, the manuscript contains some technical vocabulary. While some of it is unavoidable as it allows for precise and thus short writing, we tried to reduce it as much as possible in rewriting the manuscript. Also, we hope that the extended introduction as well as the added exhaustive discussion section help in putting the results presented in the paper in sufficient non-technical context to make them appreciable by non-experts.

Regarding results,

R3-4:

Suggestion: put the results of the analyses testing the ecological association before the ancestral reconstruction results (Fig 2), which I think are kind of complementary or descriptive and do not attempt to answer any clear specific hypotheses. A reorganization of results and rewriting to highlight the background and hypotheses being tested could make the paper more evolutionary and less descriptive, which would render it much more interesting and relevant.

A: We thank the reviewer for this suggestion. We rearranged the results and agree that this makes the structure of the paper more incremental and increases its readability.

R3-5:

pPCAs related to ecological factors explain a very low percentage of the cortical variance. Could you please expand on this? How the conclusion is granted when explained variance is so low?

A: It is true that the variance of the overall shape of the surface of the cerebral cortex explained by the selected components is very low. We attribute this to the fact that numerous other unobserved factors apart from the 3 coarse factors we described influence cortical shape. These are however both hard to collect for a large number of species and difficult to compare amongst them.

We therefore opted for a compromise in detail and availability and are clearly missing a lot of detail on the lifestyle of the individual species, overall limiting the explained variance that could be obtained. Also, our results are based solely on individual specimens for each species, and inter-individual variability is also a large factor in morphology that was infeasible to consider because of limited data availability. Nonetheless, the observed factors are statistically significant, emphasising that despite the severe limitations in data quality, information about ecological niche is contained in the shape of the cerebral cortex. In fact, the distribution of the explained variance is close to linear on a log scale of PCA components (Extended Data Figure 2, panel b), highlighting the complexity of the evolution of cortical surface shape.

However, we believe that by “decoding” the maps corresponding to the selected pPCA components, we were able to show that these correspond to meaningful modes of cortical surface morphology, as their expansion patterns are not random but rather relate (see Figure 3 and the

corresponding discussion) to neuroscientific and behavioural concepts that correspond to the respective ecological variables.

R3-6:

Regarding methods, I'm not familiar with all of them, so at this point I only have a few comments, but could potentially provide more extensive comments in a follow up review.

Minor comments:

At this point in the review process I'm not providing an extensive list of small suggestions that could be potentially changed if the authors considerably edit the manuscript, but here are a few that could help because the themes repeat all over.

R3-7:

Title: Perhaps the title is too broad. Cortical morphology in what organisms? What functional correlates?

A more specific title could help the reader quickly get an idea of what the paper will be about exactly.

A: We agree that the title was maybe too much of a "catch all" and changed it to "Evolution of Cortical Geometry and its Link to Function, Behaviour and Ecology" to emphasise the geometric nature of the results presented in the paper and the correlates it addresses. We acknowledge that this title still falls short in detailing the animals under study, but the option of adding "Primates and Rodents" is incorrect as our dataset also covers Lagomorphs, Dermopterans and Scandentians, while we fear that adding the scientifically correct term "Euarchontoglires" to the title would lead to disengagement of researchers outside of biology, most specifically neuroscience, whom we consider an important target audience.

R3-8:

Abstract

"shape of the cerebral cortex"

please be more specific. Outer shape? This should be clearly defined.

A: We changed the terminology to "surface geometry", as "By geometry of a surface one usually means characterization of its metric and curvature properties in surface curvilinear coordinates." (Altenbach 2020). While we use for the most parts Euclidean instead of curvilinear coordinates, we nonetheless believe this formulation to be the most appropriate for the intended meaning. We furthermore defined "shape" to mean "*shape of the surfaces of cerebral cortices*" at the first use of the term (Line 86) but use the shorthand "shape" in the remainder of the paper to maintain readability.

Altenbach, H., & Öchsner, A. (Eds.). (2020). Encyclopedia of continuum mechanics. Berlin, Heidelberg: Springer Berlin Heidelberg., pp. 2380

R3-9:

Line 70-73 I think this paragraph should be moved to the previous section. Also, please state clearly your research aims, some hypotheses, predictions. What and why are you doing this study? I think the introduction also does a poor job describing what is known and what is not regarding the problem the authors are tackling. This section needs expansion to clearly reflect the current knowledge and a clearly stated research question.

A: We agree with the reviewer that the original introduction was extremely compact. In the revised version of the manuscript, we substantially extended its length, hopefully giving a clearer picture of the goals and context of this study. In addition to re-arranging the ordering of the results to make them more incremental and easier to follow, we also aimed at providing context for each of them in the results section.

R3-10:

Line 74 What does the convex hull represent, or what is it? This is first mentioned here so please be more precise.

A: We added more detail to the first mention of the term “convex hull”. This section now reads:

“We computed the convex hull of these segmentations as the simplest covering of the outer surface without any folds or creases. The surface areas of the resulting as well as the original model serve as global features of the cortical geometry” (Lines 126-129)

R3-11:

**Line 79 “(...) we estimated the evolution of the area of the cortical surfaces (...)”
What does it mean to “estimate the evolution”, please be more precise.**

A: We changed this to

“We obtained a calibrated phylogeny for the analysis from the consensus tree of 100 realisations of a tip-dated Bayesian model of divergence times on which we estimated the ancestral states of the area of the cortical surfaces and their convex hulls as proxies for cortical morphology” (Lines 131-134).

More details on the phylogenetic methods used can be found in the methods section (“Phylogenetic modelling”) as well as in the supplementary methods (Section 1, “Phylogenetic modelling”).

R3-12: Line 85 “We iterated these steps, progressing upwards in the tree and upon reaching the root node resampled all surfaces to a common topology”

I don't understand what to "resample all surfaces to a common topology" means. Could you clarify?

A: "resampling to a common topology" amounts to placing all landmarks in the set of shapes at geometrically corresponding locations (as obtained from the iterative pairwise shape matching by following the phylogenetic tree). We hope that this becomes clearer by reformulating this section to

"We iterated these steps, progressing backwards in the phylogenetic tree. Upon reaching its root node, we resampled all surfaces to a common topology by placing surface vertices at geometrically corresponding locations" (Lines 140-142).

We furthermore added a figure (Figure 2) giving an overview of the processing performed on the cortical surface models that we hope elucidates these methods.

Figure 2: Methodological overview. Overview of the steps required to perform shape alignment and ancestral state reconstruction between the surface models of the cerebral cortices of two sister species. (a) For two species, anatomy and geometric features are assigned to each point on their cortical mesh; (b) after spectral embedding of both surfaces, alignment establishes dense correspondence between the cortices of the two species; (c) the shape of the common ancestors is reconstructed based on these correspondences by interpolating in the space of smooth shells; (d) finally, cortical features are estimated for the common ancestor, and the reconstruction is iterated until the root of the phylogenetic tree is reached.

More technical details of the resampling procedure can be found in both the Methods section (Section "Dataset description", Line 643ff) as well as in the supplementary methods (Section 3, "Cortical surface matching").

R3-13:

Figure 1 legend “The resulting surface models are compared 64 with endocasts (Supplementary Table 1b) obtained from phylogenetically closely related 65 fossil endocasts at the branching points indicated by the letters A-I”

There are no branching points (nodes) indicated on the phylogeny. Please correct.

A: We thank the reviewer for this observation. Embarrassingly, these labels have disappeared while generating the final figure without us noticing. We have corrected this oversight and added them back in..

R3-14:

Figure 2 – Labels are missing everywhere in the plots, it’s very hard to read. Please use letters to subdivide the figure and make it easier to reference each part in the legend.

For example “The y-axis in the plot showing the area-increase of clusters, also denotes the relative area across the cluster on the surface” could be simplified with e.g “b) area-increase of clusters. Y-axis denotes the relative area across...” or similar.

A: We agree with the reviewer that this specific figure (now Figure 7 in the revised manuscript) is difficult to interpret. As suggested, we separated it into labelled subplots for easier referencing and adapted the caption:

Figure 7: Estimated sequence of local cortical expansion from the last common ancestor of rodents and primates to *Homo*. (a) Consecutive expansion maps between the cortical surfaces are obtained from ancestral state reconstruction of all ancestors of *Homo* until the LCA with Glires. (b) Decoding these maps as correlations with term-specific statistical maps and retaining the 1% highest correlated maps at each time-point allows to perform (c) hierarchical clustering of their sequential correlation patterns in the deep ancestral human lineage (Supplementary Table 7a). (d) This reveals a decomposition of evolutionary cortical

surface expansion that can be approximated as a sequence of steps (Supplementary Table 7b), starting with the expansion of primary visual, auditory and motor regions in the Cretaceous up until the Oligocene, followed by the expansion of higher order association areas in the Miocene, Pliocene and Pleistocene. The common x-axis of all subplots a-c denotes million years ago (Mya). The colormap on the cortical surfaces (a) indicates relative area expansion between estimated speciation points. Markers for correlations in subplot (b) are jittered in horizontal direction to reduce overlap. Correlations are summarised for each cluster using violin plots in subplot (d) and joined at their median values through time. Statistical information reported in Supplementary Table 7c, License information for artwork used in Supplementary Table 7d. Abbreviations: P.I.: *Ptilocercus iowii*. N.t.: *Notharctus tenebrosus*. A.a: *Archicebus achilles*, C.: Cercopithecoidea, O.b.: *Oreopithecus bambolii*, S.t.: *Sahelanthropus tchadensis*, A.r.: *Ardipithecus ramidus*, H.s.: *Homo sapiens*.

R3-15:

Line 120 “ The maxima of association between expansion and functional categories of the components revealed a sequence of neuromorphological stages, starting with the expansion of visual areas, followed by parahippocampal regions in the late Cretaceous, auditory and sensory-motor areas in the early Oligocene until the mid-Miocene and finally expansion of association areas in the late Miocene and Pliocene up until the present day.”

Here I got lost. The section starts describing morphological changes in the “human lineage”/Homo but in the following paragraph you describe changes happening in the late Cretaceous? Primates as a whole barely existed by then. Please clarify.

A: In the original version of the manuscript, we used “human lineage” to mean the evolutionary history of humans since the last common ancestor with the evolutionary most distant species in the available dataset. We acknowledge that this terminology can easily be confused with the evolution of Homo since the divergence from Pan, so we changed the terminology throughout the text to “deep ancestral human lineage” or, as in the mentioned section, “evolutionary history of humans”.

R3-16:

Each section describes analyses but without clearly stating the purpose of such analyses. It is very hard to follow the train of thought of the authors. Please provide clear context and concise aims.

A: In response to all three reviewers, we have substantially reformatted the manuscript to make its intent and purposes clearer to the reader. While trying to not bloat the article too much, we included short context to each subsection of the results, rewrote the introduction, merged the discussion and supplementary discussion from the original submission and added a conclusion section. We have also clarified the two main aims of the paper: first, to provide evidence for the link between the evolution of cortical morphology and evolutionary adaptations to ecological niches, and second, to provide a common reference space to map cortical features across a large number of extant species to inform comparative neuroscience. We hope that these changes make the manuscript easier to grasp and more enjoyable to read.

R3-17:

Line 128 “We then compared the evolutionary expansion patterns in the two distinct lineages of mice and humans (Extended Data Figure 4) as they represent ubiquitously studied species of rodents and primates.”

Why? Just because they are very studied is not enough to justify studying them more. Again, please be more precise with the research question you are trying to answer. Guide the reader through the analyses to understand what was done and for what reason.

A: We chose to compare the evolution of the surface geometry of mice compared to humans to highlight the stark differences in the evolutionary history of the two species. Of course, other species in the dataset have evolved into even more specific niches and are thus evolutionary even more distant. However, the present manuscript is aimed not only towards comparative neuroanatomists or biologists, but rather to a general audience and importantly towards researchers in other related fields in neuroscience. There, for instance in neuropharmacology or systems neuroscience, mice represent commonly used “model” animals, many readers are familiar with their neuroanatomy and are interested in translating results obtained from the study of their brains onto humans. We therefore consider it important to highlight the important differences that exist between them and our species and how the tools published together with this manuscript can be of use in analysing and bridging these gaps.

R3-18:

Line 159 “We performed phylogenetic principal component analysis (pPCA) to test if ecological and behavioural niches are reflected in the shape of the cerebral cortex.”

PCA of what data? PCA is a dimensionality reduction technique, it’s not for testing association with other variables. You reduced the number of variables using pPCA and then tested for the association of this variables with the niche using another method. Please clarify.

A: We thank the reviewer for pointing out this imprecise and incorrect formulation. We changed the corresponding sentence to

“First, we performed phylogenetic principal component analysis (pPCA) to reduce the dimensionality of our dataset and tested for the association between the extracted modes of variation describing the shape of the cerebral cortex and if ecological and behavioural niches”
(Lines 190-192)

R3-19:

Line 170 “By mapping the relative expansion patterns into a functionally defined parcellation of the human cerebral cortex, we found significant group differences (...)”

What groups? I couldn’t follow, could you please clarify?

A: We apologise if the term “group” here is misleading. It is in fact unnecessary, the groups were simply the binary variables (arboreal / not arboreal, living in large social groups / not living in large social groups etc.). However, removing the term “group” makes the meaning of the analysis clearer as it pertains only to the effects of the ecological and behavioural variables. The section now reads

“By mapping the relative expansion patterns into a functionally defined parcellation of the human cerebral cortex, we found significant group differences (Kruskall-Wallis, $p < 0.001$, Supplementary

Table 6) in the cortical expansion patterns associated with every investigated ecological and behavioural variable except diurnality. The largest group effects were observed for ecological variables of habitat (fossorial: $\eta^2=0.21$, terrestrial: $\eta^2=0.14$, arboreal: $\eta^2=0.18$) whereas large group size showed lower group effect ($\eta^2=0.1$).

Post-hoc tests revealed significantly differing expansion patterns associated with both ecological and behavioural variables (Extended Data Figure 5).” (Line 202ff)

R3-20:

Extended Data Figure 2 – In the plots, the correct name is “Primates”, not “Primata”

What is the scale at the right of the plot showing?

A: We corrected the terminology and added a label to the colour bar (“relative cortical expansion/contraction associated with pPCA mode”):

Extended Data Figure 2: Scatter plot of total cortical surface area and gyrification index.

Performing regularised phylogenetic pPCA on a down-sampled version of the studied dataset yielded 21 potentially informative shape dimensions. **(a)** Clade membership and overall brain size are discriminated in the first 2 pPCA dimensions, while pPCA dimension 5 is associated with preferred habitat and group size. **(b)** Fraction of the variance explained in each pPCA dimension is low. We retain the first 21 modes as the cophenetic correlation between shape and the underlying genetic phylogeny drops rapidly at higher dimensions. Individual dimensions are correlated with global shape measures, but of those related to ecological or behavioural variables only pPCA dimension 6 shows significant correlation with global parameters of cortical shape. **(c)** Overall, there exists a strong negative correlation between the surface area and the gyrification of the cortex ($\rho = -0.82918$, 0.95 CI [-0.884, -0.751], $p < 0.0001$). This relationship is conserved in primates ($r = -0.92358$, 0.95 CI [-0.959, -0.859], $p < 0.0001$), but not in rodents ($r = 0.23403$ [-0.176,

0.575], $p=2.307e-01$, Supplementary Table 10a) (d) pPCA dimensions 2, 5, 6, 12 and 21 (not shown) are significantly (* $q<0.05$, ** $q<0.01$, *** $q < 0.001$, FDR-corrected, Supplementary Table 10b) related to ecological variables encoding environmental niche and group size. Boxplots represent the median with 1.5 times the inter-quartile range.

Methods

R3-21:

Line 343 “Species that were reported as either diurnal, cathemeral or arrhythmic in the literature were labelled as “diurnal”.

I’m not familiar with all studied species to provide an assessment of the validity of this simplification, but for example it is known that the platyrrhine Aotus (Night/Owl monkey) despite being “cathemeral” has extensive adaptations to this niche, for example, enlarged orbits that could potentially be reflected in cortical morphology. I’m not asking the author’s to repeat all the analyses (probably results won’t change as cathemerality in Aotus is a derived condition) but a line justifying why the simplification is granted would be useful.

A: There are of course numerous specific adaptations to specific evolutionary niches that have an influence on cortical morphology such as the large orbits of Aotus and many other specificities of the species in our dataset. We agree with the reviewer that the ecological variables we used in our study are clearly a stark oversimplification of the complex environments inhabited by each of the species as well as their resulting specific behaviours resulting - in part - from the architecture of their respective cerebral cortices. However, such simplifications are required to enable any sort of comparative analysis. In the case of diurnality, being restricted to a binary classification is especially limiting. As species living at least partly in daylight, we found the opposite choice of adding cathemeral and arrhythmic lifestyles to “nocturnal” the worse choice still, as doing so would in our opinion be more conflictive with any expected (neuro-)morphological adaptations. However, we can assure the reviewer that all classifications of the species are based on literature review (references are in the supplementary material) and were performed before any other analysis so as not to bias any results.

R3-22:

Line 508 – “However, the scalar formulation of mixture of OU models is defined only for ultrametric trees (eg. trees whose tips all have the same age). We therefore shift the tip ages for the fossil data to the present day. For the extinct hominin species included in the augmented tree, only estimates of the SA of the convex hull of the brain are available from endocasts of the fossil crania; the SA of the cerebral cortex is considered missing in phylogenetic modelling. We assume that the slight misspecification of the tip-dates for the fossil hominin data is immaterial to the relative BIC values used for model selection. “

I would argue the whole point of incorporating fossil species is that they predate extant species, so forcing them to be at the present make their use at the bare minimum pointless, and more likely, distortive of the results. Did you do this only for hominin fossils? If this is the case I could be ok with that although I see no point. But if older species were forced to be at present, this could clearly distort OU models because variance could be potentially inflated, thus impacting the model parameters (sigma, alpha and/or the optimum).

A: Only hominin fossils were added to the dataset in this way. The motivation for doing so lies in the exceptional expansion of the cerebral cortex in humans, but we agree with the reviewer that omitting this data would in all likelihood not affect the results obtained in the paper considering the range of species and the corresponding evolutionary time covered by the model. Nonetheless, we chose to keep the data - as well as the endocasts in figure 1 of the main text - to hint at the fact that comparative phylogenetic methods should always also consider the fossil record. We feel that even if the role of fossil data in the current paper is negligible, their presence is warranted by their function of subtly reminding the reader of its general importance.

Reviewers' Comments:

Reviewer #1:

Remarks to the Author:

The authors adjusted their manuscript by clarifying the goals of the paper, providing more details on the methods that are being used, and by generally improving the clarity of writing.

The goal of the paper is now much more clearly geared towards providing a resource/tool for future research. Another more empirically oriented goal is mentioned: to provide a link between cortical geometry and evolutionary adaptations. This empirical goal is, however, not new as it has been demonstrated by previous work and does not bear out of defined hypotheses (as would be expected for empirical research). There are also no discernibly new insights that stem from this goal.

Furthermore, throughout the paper the empirical goal is treated more like a validation of the resource/tool, rather than a goal in and of itself. This puts the primary emphasis of the paper directly on making available the resource/tool.

As I mentioned in my previous review, I believe that not clearly highlighting landmark new empirical insights limits the impact of the work. In response to my comment, the authors seem to have doubled down on stating that the goal of this paper is to provide a new resource/tool, not to provide new empirical insights. In general, my perspective is that a resource/tool is best exemplified by showing which new insights it can provide. Providing the resource and leaving potential new insights for future research changes the scope of this paper towards being a methodological paper.

Overall, I find that the manuscript is well improved in terms of clarity. The scope, however, is now clearly geared towards providing a resource/tool for future research. It seems that perhaps it may fit better within the aims and scope of Nature Methods.

Aligning with this, it seems that the title is more appropriately captured by the heading of the first section of the results ("An evolutionary common reference frame of cortical geometry for comparative neuroscience"). I understand that the authors already changed their title, but the current title suggests that the paper is empirically oriented and implies it will provide new insights into the "evolution of cortical geometry and its link to function, behavior and ecology". I can understand it may be frustrating to authors for reviewers to continue suggesting different titles, but this stems primarily from the lack of clarity in the goal of the paper. As the authors have clarified the goal of their paper, it has become clearer that the primary goal is to provide a resource. The title should reflect this. In a way, the abstract already reflects this by not providing a clear new empirical insight but emphasizing how the joint reference space may facilitate future investigations. An empirical title followed by a methodological paper does not fit.

I remain convinced that this work is impressive and deserves to be highlighted in a high-impact journal. At the same time, with the 'new' goal of providing a resource that is heavily weighted towards geometric morphometric methodology, I believe this paper has gone in a direction that is beyond my expertise. As I mentioned in my previous review, my expertise lies with brain evolution and phylogenetic comparative methodology, not with geometric morphometric methodology.

Reviewer #3:

Remarks to the Author:

This is the revised version of a manuscript reconstructing the brain's evolutionary history across Euarchoontoglyres species.

I found this version to be excellent and basically ready for publication. Compared to the previous version which was incredibly hard to understand and follow, I really enjoyed reading this revised version. It flows naturally and although some of the methods are beyond my expertise it is relatively easy to grasp the ideas being tested and how it was done. So I commend the authors for the effort and the final result. The amount of information, data and new hypotheses generated by this work is massive and it will undoubtedly be very influential for the field.

So I basically have no further comments as I see no fatal flaws. I think at this point is no longer my

task to give my opinion on the results and its time for the readers to decide.

Reviewer #4:

Remarks to the Author:

The first version of this manuscript had three major issues which concerned reviewers. First, the introduction was confusing and as a result the aims of the paper were difficult to follow. Secondly, the methods were not explained in sufficient detail. Third, some of the discussion suffered from the same problems as the introduction: too much focus on features not tested by the study (function/cognition) and some concerns about how the authors assessed different locomotory lifestyles.

Regarding #1, the authors have made significant improvements to the clarity and presentation of the scientific questions. They present the opposing models of brain evolution clearly and removed unnecessary speculation on cognition.

Re: #2, the creation of a new figure 2, providing a graphical overview of the methods, is a welcome addition that will help readers follow the procedures. Further, they expanded their descriptions of the creation of common reference space, which is a critical part of the methods that was not included in the main text earlier.

Re: #3, the Discussion is greatly improved by including a more precise discussion of mammalian adaptations and their putative relationship to anatomical differences/changes, as well as a more nuanced approach to discussing evolutionary processes.